# Documenting numerical experiments in support of CMIP6

Charlotte Pascoe[1,2], Bryan N Lawrence[2,3], Eric Guilyardi[2,4], Martin Juckes[1,2], and Karl E Taylor[5]

[1]STFC Rutherford Appleton Laboratory, UK.
[2]National Centre of Atmospheric Science, UK.
[3]Departments of Meteorology and Computer Science, University of Reading, UK.
[4]LOCEAN/IPSL, Sorbonne Université/CNRS/IRD/MNHN, Paris, France
[5]PCMDI, Lawrence Livermore National Laboratory, Livermore, CA, USA

*Correspondence to:* Charlotte Pascoe (charlotte.pascoe@ncas.ac.uk)

**Abstract.** Numerical simulation, and in particular simulation of the earth system, relies on contributions from diverse communities, from those who develop models to those involved in devising, executing, and analysing numerical experiments. Often these people work in different institutions and may be working with significant separation in time (particular analysts, who may be working on data produced years earlier), and they typically communicate via published information (whether journal papers, technical notes, or websites). The complexity of the models, experiments, and methodologies, along with the diversity (and sometimes inexact nature) of information sources, can easily lead to misinterpretation of what was actually intended or done. In this paper we introduce a taxonomy of terms for more clearly defining numerical experiments, put it in the context of previous work on experimental ontologies, and describe how we have used it to document the experiments of the sixth phase for the Coupled Model Intercomparison Project (CMIP6). We describe how through iteration with a range of CMIP6 stakeholders we rationalised multiple sources of information and improved the clarity of experimental definitions. We demonstrate how this process has added value to CMIP6 itself by a) helping those devising experiments to be clear about their goals and their implementation, b) making it easier for those executing experiments to know what is intended, c) exposing inter-relationships between experiments, and d) making it clearer for third parties (data users) to understand the CMIP6 experiments. We conclude with some lessons learned, and how these may be applied to future CMIP phases as well as other modelling campaigns.

## 1  Introduction

Climate modelling involves the use of models to carry out simulations of the real world, usually as part of an experiment aimed at understanding processes, testing hypotheses, or projecting some future climate system behaviour. Executing such simulations requires an explicit understanding of experiment definitions including knowledge of how the model must be configured to correctly execute the experiment. This is often not trivial, especially when those executing the simulation were not party to the discussions defining the experiment. Analysing simulation data also requires at least minimal knowledge of both the models used and the experimental protocol to avoid drawing inappropriate conclusions. This again can be non-trivial, especially when the analysts are not close to those who designed and/or ran the experiments.

Traditionally numerical experiment protocols have appeared in the published literature, often alongside analysis. This approach has worked for years, since mostly the same individuals designed the experiment, ran the simulations, and carried out the analysis. However, as model inter-comparison has become more germane to the science, there has been growing separation between designers, executers, and analysts. This separation has become acute with the advent of sixth Coupled Model Inter-comparison Project (CMIP6, Eyring et al., 2016). With dozens of models and experiments, dozens of modelling centres engaged, and hundreds of output variables, it is no longer possible for all modellers to fully digest all the nuances of all the experiments which they are required to execute. Simulations are now carried out for direct application within specific Model Inter-comparison Projects (or MIPs), for re-use between MIPs, and often with an explicit requirement that they be made available to support serendipitous analysis. Much of such re-use is by people who have no intimate knowledge of either the model or the experiment.

This increasing separation within the workflow, and between individuals and communities, leads to an increased necessity for information transfer, both between people and across time (often analysts are working years after those who designed the experiments have moved on). In this paper we introduce the "design" component of the ES-DOC ontology, intended to aid in this information transfer by supporting both those designing experiments (especially those with inter-experiment dependencies) and those who try to execute and/or understand what has been executed. This ontology provides a structure and vocabulary for building experiment descriptions which can be easily viewed, shared, and understood. It is not intended to supplant journal articles, rather to provide recipes which can be re-used (by those running models) and understood by analysts as an introduction to the experiment designs. We explain how it was deployed in support of CMIP6, how it has added value to the CMIP6 process, and how we expect it to be used in the future based on lessons learned thus far.

We begin by describing key elements of simulation workflows and introduce a formal vocabulary for describing the experiments and the simulations. We provide some examples of ES-DOC compliant experiment descriptions, and then present some of the experiment linkages which can be understood from the use of our canonical experiment descriptions. Our experiences in gathering information and the linkages (and some of the missing links) required to define and document CMIP6 experiments expose opportunities for improving future MIP designs, which we present in the summary.

## 2 Structured Experiment Documentation

In this section we introduce the key concepts involved in designing experiments and describing simulation workflows. We describe how this has evolved from previous work and differs from other work with which we are familiar.

### 2.1 Experiment Definition

The process of defining numerical experiments is potentially complex (Fig. 1(a)). It begins with an idea and often entails an iterative community discussion which results in the final experimental definition and documentation. In the simplest cases, such documentation may be prose, in a manuscript or a journal article, but when many detailed requirements are in play and/or many experiments and individuals are involved, it is helpful to structure the documentation - both to ensure that key steps

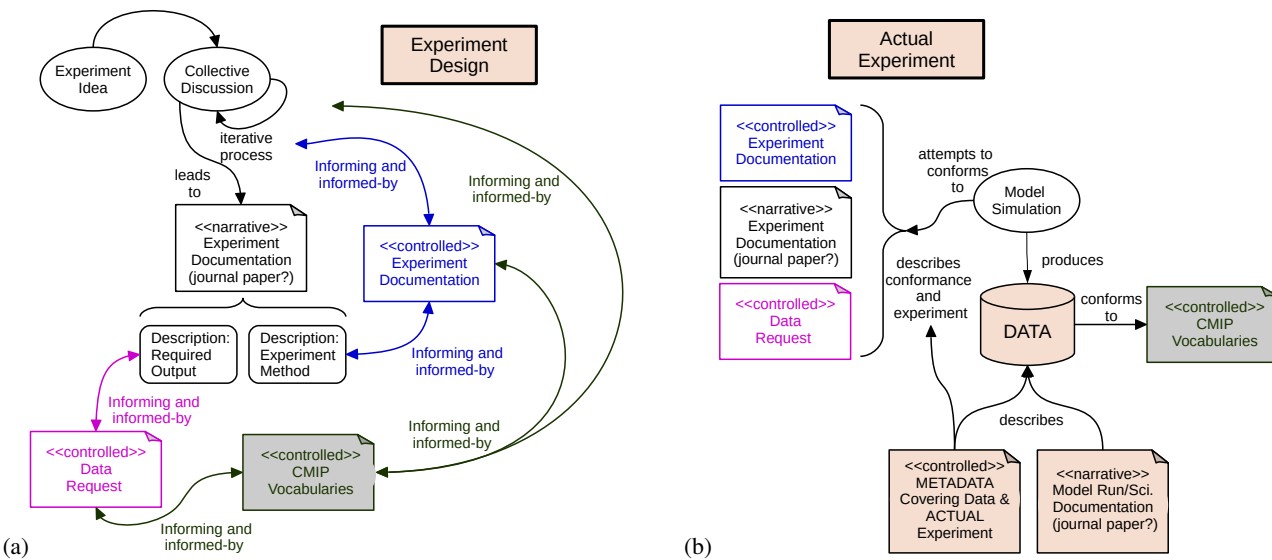

**Figure 1.** (a) The process of defining an "experiment" involves multiple steps, interactions, and component descriptions. In the simplest case, ideas are iterated leading to some sort of final description (white boxes), but at scale, there is a need to control the structure used to document the experiments (blue) and their intended output (data request, magenta), and such structure needs to utilise controlled vocabularies (shaded). (b) The realisation of an experiment is carried out by a model simulation which produces data, but in practice simulations often deviate in detail from the experiment protocol, and such deviations themselves need to be recognisable; how well a simulation conforms to the protocol is a key element of the documentation. In both (a) and (b) the «narrative» and «controlled» notation indicates the key characteristics of the two types of documentation: the former in scientific prose for human readers, the latter more structured for consumption both by humans and automated machinery.

are recorded, and to aid in the inter-comparison of methodology between experiments (especially the automatic generation of tables and views). Key requirements include being very specific about imposed experimental conditions and the required output.

Once the experiments are defined (Fig. 1(a)), modelling groups realise the experiments in the form of simulations which
5    attempt to conform to the specifications of the experiment and which produce the desired output (Fig. 1(b)).

In both generic experiment documentation, and in defining data requests, it is helpful to utilise controlled vocabularies so that unambiguous machine navigable links can exist between the design documentation, simulation execution, data production, and the analysis outcomes.

## 2.2   Key concepts

10   The requisite controlled vocabulary for a numerical simulation workflow requires addressing the the actions and artefacts of the workflow summarised in Fig. 2, in which we see *Projects* (e.g., MIPs) design *Numerical Experiments* and define their *NumericalRequirements*. (In this section we use italics to denote specific concepts in the ES-DOC taxonomy.) Experiment definitions

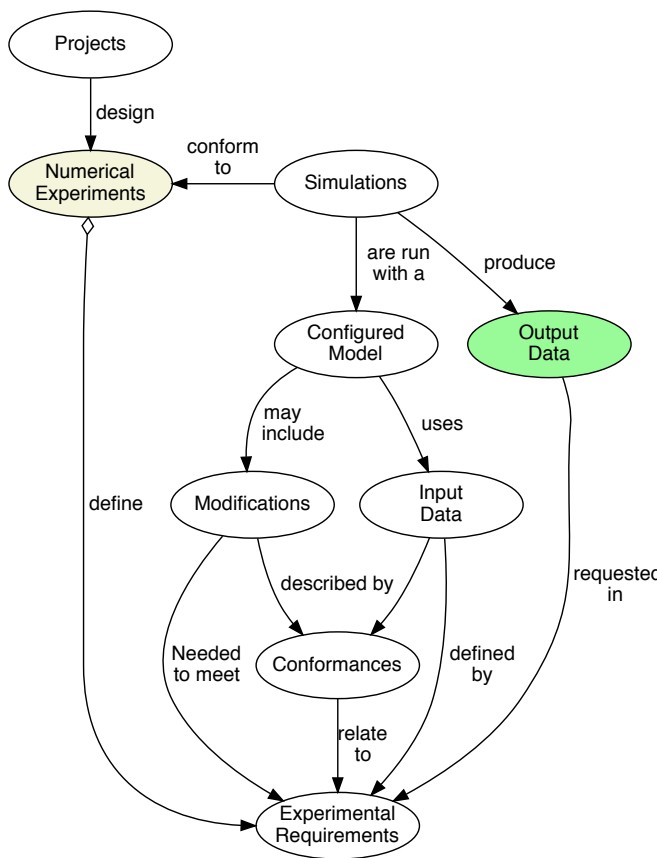

**Figure 2.** Simulation workflow in which experimental requirements (termed "Numerical Requirements") play a central role.

are adopted by modelling groups who use a model to run *Simulations*, with *Output Data* requirements ("data requests") being one of the many experimental requirements. A simulation is run with a *Configured Model*, using a configuration which will include details of *InputData* and may include *Modifications* required to conform to the experiment requirements. Not all of the configuration will be related to the experiment, aspects of the workflow and computing environment may also need to be

5  configured. In practice, simulations can deviate in detail from the experiment protocol; that is, they do not conform exactly to the requirements. A key part of a simulation description, then, is the set of *Conformance* descriptions which indicate how the simulation conforms to the experimental requirements. In this paper we are limiting our attention to the definition of the Experiment and its Requirements, with application to CMIP6 and the relationship between the MIPs and those requirements. We address other parts of the workflow elsewhere.

10  As noted above, a Project has certain scientific objectives that lead it to define one or more *NumericalExperiment*s. We describe the rules for performing the numerical experiments as *NumericalRequirement*s (Fig. 3). Both *NumericalExperiment*s and *NumericalRequirement*s may be nested and the former may also explicitly identify specific related experiments which may provide dependencies or other scientific context such as heritage. For example, an experiment from which initialisation fields

are obtained is referred to as the parent experiment. Nested requirements are used to bundle requirements together for easy re-use across experiments. (An example of a nested requirement can be seen in the appendix where Table A1 shows how all the components which go into a common CMIP6 pre-industrial solar particle forcing are bundled together. We will see later that in CMIP6, many implicit relationships arise from common requirements.)

5   The experiment description itself includes attributes covering the scientific objective and the experiment rationale addressing the questions: What is this experiment for? Why is it being done?

## 2.3   Requirements

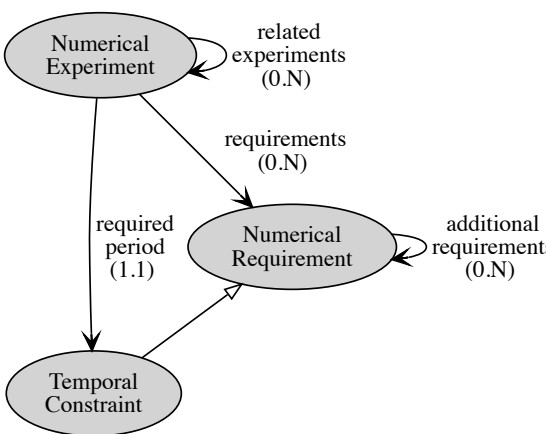

**Figure 3.** *Numerical Experiment*s are designed and governed by MIPs. Each numerical experiment is defined by *Numerical Requirements*, including a mandatory constraint setting out the required period of the numerical experiment. Numerical requirements may have complicated internal structures (see fig4). In both this and the next figure, arrows and their labels use the Unified Modelling Language (UML) syntax to describe the relationships between the entities named in the bubbles. UML provides a standard way to visualise the components of a system and how they relate to each other, different styles of arrow denote different types of relationship. The UML relationships used here and in Fig. 4 are described in Table 1 of Hassell et al. 2017, a short primer to UML concepts can be found in Appendix A of Hassell et al. 2017.

The *NumericalRequirement*s are the set of instructions required to configure a model and provide prescribed input needed to execute a simulation that conforms to a *NumericalExperiment*. These instructions include (Fig. 4) specifications such as the

10   start date, simulation period, ensemble size and structure (if required), any forcings (e.g. external boundary conditions such as the requirement to impose a one percent increase in carbon dioxide over 100 years), initialisation requirements (e.g. whether the model should be "spun-up" or initialised from the output of a simulation from another experiment), and domain requirements (for limited area models). A scope keyword from a controlled vocabulary can be used to indicate whether the requirement is re-used elsewhere, e.g. in the specifications for related experiments.

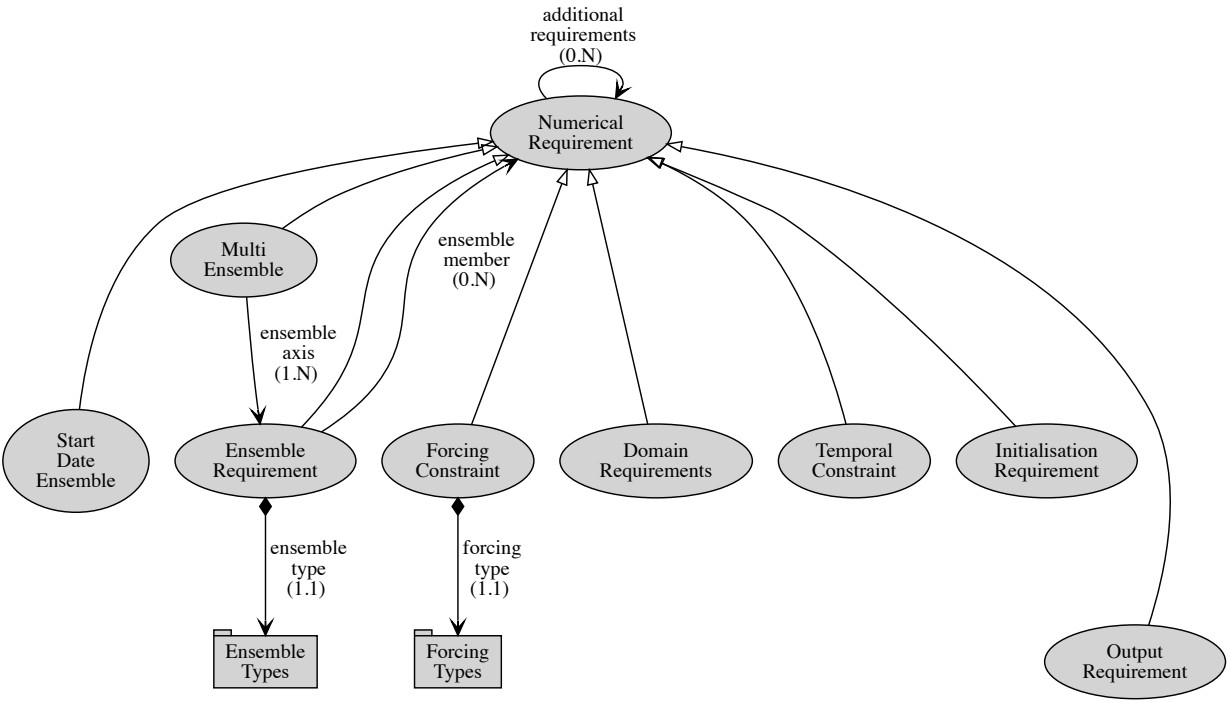

**Figure 4.** *NumericalRequirements* govern the structure of a numerical experiment covering constraints on duration (*TemporalConstraint*), the domain covered (*DomainRequirement*, e.g. global or a regional bounding box), any forcings (*ForcingConstraint*, such as particular greenhouse gas concentrations), output requirements (e.g. the CMIP6 data request), and a complicated interplay of potential *EnsembleRequirements* (see text). Controlled vocabularies are necessary for *EnsembleTypes*, *ForcingTypes*, and *NumericalRequirementScopes*. Indices associated with the connectors indicate the numerical nature of the relationships e.g. a *NumericalRequirement* can have anywhere between zero to many (0.N) additional requirements whereas an *EnsembleRequirement* can have only one (1.1) *EnsembleType*.

| Forcing Constraint | | | |
|---|---|---|---|
| **attribute** | **type** | **cardinality** | **description** |
| category | str | 0.1 | Category to which this belongs (from a CV, e.g. GASES). |
| code | str | 0.1 | Programme wide code from a controlled vocabulary (e.g. N2O). |
| data_link | *data.dataset* | 0.1 | A data record used by the forcing |
| forcing_type | *designing.forcing_types* | 1.1 | Type of integration |
| group | str | 0.1 | Sub-Category (e.g. GHG) |
| origin | *shared.citation* | 0.1 | Pointer to origin, e.g. CMIP6 RCP database. |

**Table 1.** ES-DOC controlled structure for describing a forcing constraint: each attribute has a name, a python data type (those in italics are other ES-DOC types), a cardinality (0.1 means either zero or one, 1.1 means one is required) and a description.

| Forcing Types | |
|---|---|
| **keyword** | **definition** |
| historical | Best estimates of actual state (included synthesised) |
| idealised | Simplified and/or exemplar, e.g. 1%C02 |
| scenario | Intended to represent a possible future, e.g. RCP4.5 |
| driven | Driven with data output from another simulation |

**Table 2.** ES-DOC Forcing types controlled vocabulary, provides context for a forcing constraint.

Each requirement carries a number of optional attributes and may contain mandatory attributes, as shown in Tables 1 and 2 for a *ForcingConstraint*.

## 2.4 Related Work

The ES-DOC vocabulary is an evolution of the "Metafor" system (Lawrence et al., 2012; Guilyardi et al., 2013; Moine et al., 2014), developed to support the fifth CMIP phase (CMIP5). Metafor was intended to provide the structured vocabulary and tools to allow those contributing simulations to CMIP5 to document their models and simulations. In that context, Metafor was a qualified success; useful information was collected, but the tools were not able to be fully tested before use and were found to be difficult to use by those providing the documentation content. Such difficulties resulted in documentation generally arriving too late to be of use to the target audience: scientists analysing the data. The lessons learned from that exercise were baked into the ES-DOC project which has superseded Metafor, leading to a much improved ontology, better tooling, and improved viewing of the resulting documentation (https://es-doc.org).

The ES-DOC controlled vocabulary is an instance of an ontology ("a formal specification of a shared conceptualisation", Borst 1997). There is considerable literature outlining the importance of such ontologies in establishing common workflow patterns with the goal of improving reproduction of results and reuse of techniques (whether they be traditional laboratory

experiments or *in silico*) and explicitly calling out the failure of published papers as a medium to provide all the details of experiment requirements (e.g. Vanschoren et al. 2012 in the context of reproducible machine learning).

The description of ontologies is often presented in the context of establishing provenance for specific workflows and often only retrospectively. Work supporting scientific workflows has mainly been concerned with execution and analysis phases, with little attention paid to the composition phase of workflows (Mattoso et al., 2010), let alone the more abstract goals.

For the "conception phase" of workflow design, a controlled vocabulary introduced by Mattoso et al. as part of their proposed description of "experiment life cycles" directly maps to our work on experiment descriptions (discussed in this paper). In their view, the conception phase potentially consists of an abstract workflow, describing what should be done (but without specifying how), and a concrete workflow, binding abstract workflows to specific resources (models, algorithms, platforms, etc). ES-DOC respects that split with an explicit separation of design (experiment descriptions) and simulation (the act of using a configured model in an attempt to produce data conforming to the constraints of an experiment).

The notion of "an experiment" also needs attention, since the experiments described here are even more abstract than the notion of "a workflow" and cover a wider scope than that often attributed to an experiment. Dictionary definitions of "scientific experiment" generally emphasise the relationship between hypothesis and experiment (e.g. "An experiment is a procedure carried out to support, refute, or validate a hypothesis. Experiments provide insight into cause-and-effect by demonstrating what outcome occurs when a particular factor is manipulated.", Wikipedia contributors 2018). In this context "factor" has a special meaning, a factor generally being one of a few input variables; but in numerical modelling there can be a multiplicity of such factors, leading to difficulties in formal experimental definition and consistency of results (Zocholl et al. 2018 in the context of big data experiments).

The first formal attempt to define a generic ontology of experiments (as opposed to workflows), appears to be that of Solda-tova and King (2007) (who also expressly identify the limitations of natural language alone for precision and disambiguation). Key components of their ontology include the notions of experimental classification, design, results, and their relationships, but it is not obvious how this ontology can be used to guide either conception or implementation. da Cruz et al. (2012) build on Mattoso et al. (2010) to specify more fully the abstract conception phase of workflow with more generic experiment concepts with much the same aim as Soldatova and King, however, they introduce many elements in common with ES-DOC and one could imagine some future mapping between these ontologies (although there is not yet any clear use case for this).

With the advent of simulation, another type of experiment (beyond those defined earlier) is possible: the simulation (and analysis) of events which cannot be measured empirically, such as predictions of the state of a system influenced by factors which cannot be replicated (or which may be hypothetical, such as the climate on a planet with no continents). For climate science, the most important of these is of course the future; experiments can be used to predict possible futures (scenarios).

In this form of experiment, ES-DOC implicitly defines two classes of "controllable factor": those controlled by the experiment design (and defined in *NumericalRequirement*s, in particular, by constraints) and those which are controlled by experiment implementation (the actual modelling system). Only the former are discussed here. Possibly because most of the existing work does not directly address this class of experiment, there is no similar clear split along these lines in the literature we have seen.

## 3 CMIP6

The rationale and need for CMIP6 were introduced in Meehl et al. (2014), and the initial set of MIPs which arose are documented in Eyring et al. (2016). In this section we discuss a little of the history leading to CMIP6 in terms of how the documentation requirement has evolved, we discuss the interaction of various players in the specification of the experiments
and how that has led to the ES-DOC descriptions of the CMIP6 experiments and their important forcing constraints.

### 3.1 History

Global model intercomparison projects have a long history, with pioneering efforts beginning in the late 1980's (e.g., Cess et al. 1989 and Gates et al. 1999). The first phase of CMIP was initiated in the mid 1990s (Meehl et al., 1997). CMIP1 involved only a handful of modelling groups, but participation grew with each succeeding phase of CMIP. Phase six (CMIP6, underway
now) will involve dozens of institutions, including all the major climate modelling centres and many smaller modelling groups. Throughout the CMIP history, there has been a heavy reliance on CMIP results in the preparation of Intergovernmental Panel on Climate Change (IPCC) reports — CMIP1 diagnostics were linked to IPCC diagnostics and the timing of CMIP phases has been associated with the IPCC timelines.

With each phase, more complexity has been introduced. CMIP1 had four relatively simple goals: to investigate differences in
the models' response to increasing atmospheric CO2, to document mean model climate errors, to assess the ability of models to simulate variability, and to assess flux adjustment (Sausen et al., 1988). CMIP6 continues to address the first three of these objectives (flux adjustment being rarely used in modern models), but with a broader emphasis on past, present and future climate in a variety of contexts covering process understanding, suitability for impacts and adaptation, and climate change mitigation.

In CMIP5 and again in CMIP6, there was a substantial increase in the number and scope of experiments. This has led to a new organisational framework in CMIP6 involving the distributed management of a collection of quasi-independently constructed Model Intercomparison Projects, which were required to meet requirements and expectations set by the overall coordinating body (the CMIP Panel) before they were "endorsed" as part of CMIP6. These MIPS were designed in the context of both increasing scope and wider-spread interest, and the growth of two important constituencies: (1) those designing "Diagnostic
MIPs", which do not require new experiments, but rather request specific output from existing planned experiments to address specific interests; and (2) the even wider group of downstream users who use the CMIP data opportunistically, having little or no direct contact with either the MIP designers or the modelling groups who ran the experiments.

With the increasing complexity, size, and scope of CMIP came a requirement to improve the documentation of the activity, from experiment specification to data output. CMIP5 addressed this in three ways: by documenting the experiment design in
a detailed specification paper (Taylor et al., 2011); by improving documentation of metadata requirements and data layout to improve access to, and interpretation of, simulation output; and by requiring model participants to exploit the Metafor system (section 2.4) to describe their models and simulations. ES-DOC use is now required for the documentation of CMIP6 models and ensembles (Balaji et al., 2018).

## 3.2 Documentation and the MIP Design Process

The overview of the experiment design process given in Fig. 1 can be directly applied to the way many of the CMIP6 MIPs were designed. For CMIP6 the iterative process involved the CMIP Panel[1], the CMIP6-Endorsed MIPs, the CMIP team at the Program for Climate Model Diagnosis and Intercomparison (PCMDI[2]), the ES-DOC team, and the development of the data request (Juckes et al., 2019). The discussion revolved around interpreting and clarifying the MIP requirements in terms of data and experiment set up, as initially described by endorsed MIP leaders in their proposals to the CMIP panel and later in a special issue of Geophysical Model Development (GMD)[3]. The ES-DOC community worked towards additional precision in the experiment decisions (in accordance with the structure described in section 2.2) and sought opportunities for synergy between MIPS. The CMIP6 team at PCMDI developed the necessary common controlled cross-experimental CMIP vocabularies (the CMIP6-CV). The data request was an integral part of the process, since some MIPs were dependent on data produced in other MIPs, and in all cases the data was the key interface between the aspirations of the MIP and the community of analysts who need to deliver the science.

The semantic structure of the data request was developed in parallel to the development of the CMIP6 version of ES-DOC each had to deal with a distinctive range of complex expectations and requirements. Hence ES-DOC has not yet fully defined or populated the *OutputRequirement* shown in Fig. 4. Similarly, the Data Request was not able to fully exploit ES-DOC experiment descriptions. A future development will bring these together, and make use of the relationships between MIPs and between their output requirements and objectives. However, despite some semantic differences, there was communication between all parties throughout the definition phase.

The initial ES-DOC documentation was generated from a range of sources and then iterated with (potentially) all parties involved, which provided both challenges and opportunities. An example of the challenge was keeping track of material through changing nomenclature. Experiment names were changed, experiments were discarded, and new experiments were added. In one case an experiment ensemble was formed from a set of hitherto separate experiments. Conversely, a key opportunity was the ability to influence MIP design to add focus and clarity, including influencing those very names. For example, the names of experiments which applied SST anomalies for positive and negative phases of ocean oscillation states were changed from "plus" to "pos" and "minus" to "neg" to better reflect the nature of the forcing and the relationship between experiment objectives and names.

The ES-DOC documentation process also raised a number of discrepancies and duplications, which were sorted out by conversations mediated by PCMDI. Many of the latter arose from independent development within MIPs of what eventually became shared experiments between those MIPs. For example, not all shared experiment opportunities were identified as such by the MIP teams, and it was the iterative process and the consolidated ES-DOC information which exposed the potential for shared experimental design (and significant savings in computational resources).

---

[1]https://www.wcrp-climate.org/wgcm-cmip/cmip-panel

[2]https://pcmdi.llnl.gov/CMIP6/

[3]https://www.geosci-model-dev.net/special_issue590.html

A specific example of such a saving occurred with ScenarioMIP and CDRMIP, which both included climate change overshoot scenario experiments that examine the influence of $CO_2$ removal (negative net emissions) from 2040-2100 following unmitigated baseline scenarios through to 2040. As originally conceived, the ScenarioMIP experiment (ssp534-over) utilised year 2040 from the CMIP6 updated RCP8.5 for initialisation, but the CDRMIP equvalent (esm-ssp534-over) requested initial-
isation in 2015 fom the esm-historical experiment. In developing the ES-DOC descriptions of these experiment it was apparent that CDRMIP could follow the ScenarioMIP example and initialise from the C4MIP experiment esm-ssp585 in 2040 and avoid 25 years of unnecessary simulation (by multiple groups). This is now the recommended protocol.

Discrepancies also arose from the parallel nature of the workflow. For example, specifications could vary between what was published in a CMIP6-Endorsed MIP's GMD paper and what had been agreed by the MIP authors with the Data Request and/or
the PCMDI team with the controlled vocabulary. On occasion ES-DOC publication exposed such issues, resulting in revisions all round. This process required the sustained attention of representatives of each of these groups and eventually resulted in a system relying on Slack[4] to notify all involved of updates, but usually requiring initiation by a human who has identified an issue. However, synchronicity was and is a problem, with quite different timescales involved in each of the processes. For example, the formal literature itself evolved and so version control has been important — all current ES-DOC documents cite the
literature as it was during the design phase, and will be updated as necessary. A rather late addition to the taxonomies supported by both ES-DOC and PCMDI was support for aliases, to try and minimise issues arising from parallel naming conventions for experiments. The use of aliases addressed the documentation and specification issues associated with experiment names evolving or being specified differently within a MIP and the wider CMIP6. For example some GeoMIP experiments have very different names in the GeoMIP GMD paper and in the CMIP6-CV e.g. "G1extSlice1" vs "piSST-4xCO2-solar".

This process had other outcomes too: LUMIP originally had a set of experiments that were envisaged to address the impact of particular behaviours such as "grass crops with human fire management". Some of these morphed to become entirely the opposite of their original incarnation, such as "land-noFIRE" where the experiment requires no human fire management (see Table A2). Rather than building experiments that simulate the effect of including a phenomena, the LUMIP constrained this suite of experiments in term of the phenomena that were removed from the model. This change prompted a discussion about
how then to describe experiments that are built around the concept of missing out one or more process. For instance, with a suite of experiments that require that the land scheme is run without phenomena A, then without phenomena B, without phenomena C and finally without phenomena D, can we define the individual experiments in the suite with the form "not A but with B, C and D" and "not B but with A, C and D", as in the case of an experiment where one forcing constraint might be set to pre-industrial levels whilst the rest of the forcing constraints are set to present day conditions? It turns out that there isn't
yet much uniformity about how land models are set up; each is very different, so it only makes sense for LUMIP to constrain this suite of experiments in terms of the phenomena that is removed. That is, the experiments should simply be described with the anti-pattern "not A" and "not B". It has become clear that the way an experiment's forcing constraints are framed depends to some extent on the maturity and uniformity of the models that are expected to run the simulations.

---

[4]https://slack.com/

### 3.3 Forcing Constraints in Practice

Somewhat naively, the initial concepts for *ForcingConstraints* anticipated the description of forcing in terms of specific input boundary conditions or, perhaps, specific modifications needed to models — this was how they were described for the CMIP5 documentation. The ES-DOC semantics introduced for CMIP6 are more inclusive and allowed a wider range of possible forcing constraints. For example, in CMIP5 the infamous Metafor questionnaire asked modellers to describe how they implemented solar forcing. In CMIP6, the approach to solar forcing requirements was outlined in the literature (Matthes et al., 2017), and the resulting requirements are found in rather more precise forcing constraints (with additional related requirements), an example of which appears in Table A1. The ES-DOC documentation now provides a checklist of important requirements and a route to the literature for both those implementing the experiments and those interpreting their results. Modellers can now use this information both in setting up their simulations and in documenting that setup. A discussion of how the latter is done for CMIP6 (and how it builds on lessons learned from the generally poor experiences interacting with an excessively long and complicated CMIP5 questionnaire) will appear elsewhere.

Increasing precision is evident throughout CMIP6 and in the documentation. In some cases, rather than ask how it is done in a model post fact, the experiment definition describes what is expected, as in the GeoMIP experiment G7SST1-cirrus (Table A3) where explicit modelling instructions are provided. However, where appropriate, experiments still leave it open to modelling groups to choose their own methods of implementing constraints, e.g. the reduction in aerosol forcing described in GeoMIP experiment G6sulfur (Table A4).

## 4 Experiment Relationships

CMIP6 is more than just an assemblage of unrelated MIPs. One of the beneficial outcomes of the formal documentation of CMIP6 within ES-DOC has been a clearer understanding of the dependencies of MIPs on each other, and of experiments on shared forcing constraints. In this section we provide an ES-DOC generated overview of CMIP6, discuss elements of commonality, and how these interact with the burden on modellers of documenting how their simulation conformed (or not) to the experiment requirements.

### 4.1 An overview of CMIP6 via ES-DOC

At the heart of the current CMIP process is a central suite of experiments known as the DECK (Diagnosis, Evaluation, and Characterization of Klima (Eyring et al., 2016). The DECK includes a pre-industrial control under 1850 conditions, an atmosphere-only AMIP simulation with imposed historical sea surface temperatures, and two idealised $CO_2$ forcing experiments where in one $CO_2$ is increased by 1 percent per year until reaching four times the original concentration, while in the other $CO_2$ is abruptly increased to four times the original concentration. Variants of most of these fundamental experiments have been core to CMIP since the beginning, and now within the DECK there is a second variant of the pre-industrial control designed to test the relatively new earth system models which respond to internally calculated $CO_2$ concentrations as opposed to responding to

| DECK (CMIP6) | |
|---|---|
| *Diagnosis, Evaluation, and Characterization of Klima (Climate)* | |
| **Description:** Core simulations for climate model intercomparison. | |
| **Rationale:** To maintain continuity and help document basic characteristics of models across different phases of CMIP. To investigate differences in the model's response to increasing atmospheric CO2. | |
| **Experiments** | |
| **esm-piControl**: A pre-industrial control simulation with non-evolving pre-industrial conditions and atmospheric CO2 calculated. Conditions chosen to be representative of the period prior to the onset of large-scale industrialization, with 1850 being the reference year. The piControl starts after an initial climate spin-up, during which the climate begins to come into balance with the forcing. The recommended minimum length for the piControl is 500 years. To be performed with an Earth System Model (ESM) that can calculate atmospheric CO2 concentration and account for the fluxes of CO2 between the atmosphere, the ocean, and biosphere. | **esm-piControl-spinup**: A pre-industrial control spin-up simulation with non-evolving pre-industrial forcing and atmospheric CO2 calculated. Conditions chosen to be representative of the period prior to the onset of large-scale industrialization, with 1850 being the reference year. This experiment describes an initial climate spin-up, during which the climate begins to come into balance with the forcing. To be performed with an Earth System Model (ESM) that can calculate atmospheric CO2 concentration and account for the fluxes of CO2 between the atmosphere, the ocean, and biosphere. Run until Earth System reaches equilibrium. |
| **piControl-spinup**: A pre-industrial control spin-up simulation with non-evolving pre-industrial forcing. Forcing conditions are chosen to be representative of the period prior to the onset of large-scale industrialization, with 1850 being the reference year. This experiment describes an initial climate spin-up, during which the climate begins to come into balance with the forcing. Run until at least the surface climate reaches equilibrium. | **piControl**: A pre-industrial control simulation with non-evolving pre-industrial conditions. Conditions chosen to be representative of the period prior to the onset of large-scale industrialization, with 1850 being the reference year. The piControl starts after an initial climate spin-up, during which the climate begins to come into balance with the forcing. The recommended minimum length for the piControl is 500 years. |
| **1pctCO2**: Increase atmospheric CO2 concentration gradually at a rate of 1 percent per year. The concentration of atmospheric carbon dioxide is increased from the global annual mean 1850 value until quadrupling. | **amip**: An atmosphere only climate simulation using prescribed sea surface temperature and sea ice concentrations but with other conditions as in the Historical simulation. |
| **abrupt-4xCO2**: Impose an instantaneous quadrupling of the concentration of atmospheric carbon dioxide from the global annual mean 1850 value, then hold fixed. | |

**Table 3.** The experiments within the DECK, as described in ES-DOC. The content of this table, like all the ES-DOC tables in this paper, was generated directly from the online documentation using a python script (details in the appendix). The choice of content to display was made in the python code; other choices could be made (e.g., see https://documentation.es-doc.org/cmip6/mips/deck).

externally imposed CO2 concentration (Table 3). Completion of the suite of DECK experiments is intended to serve as an entry card for model participation in the CMIP exercise. The CMIP panel are responsible for DECK design and definition, which should evolve only slowly over future phases of CMIP and will enable cross-generational model comparisons. CMIP is also responsible for the "historical" experiments, but the definition of these will change as better forcing data becomes available

5    and as the historical period extends forward in time.

| CMIP6 (core MIPS recorded by ES-DOC) |
|---|
| **AerChemMIP**: Aerosols and Chemistry MIP - Collins et al. (2016) |
| **C4MIP**: Coupled Climate Carbon Cycle MIP - Jones et al. (2016) |
| **CDRMIP**: The Carbon Dioxide Removal Model Intercomparison Project - Keller et al. (2018) |
| **CFMIP**: Cloud Feedback Model Intercomparison Project - Webb et al. (2017) |
| **CMIP**: Climate Model Intercomparison Project - Eyring et al. (2016) |
| **DAMIP**: Detection and Attribution Model Intercomparison Project - Gillett et al. (2016) |
| **DCPP**: Decadal Climate Prediction Project - Boer et al. (2016) |
| **FAFMIP**: Flux-Anomaly-Forced Model Intercomparison Project - Gregory et al. (2016) |
| **GMMIP**: Global Monsoons Modeling Inter-comparison Project - Zhou et al. (2016) |
| **GeoMIP**: The Geoengineering Model intercomparison Project - Kravitz et al. (2015) |
| **HighResMIP**: High Resolution Model Intercomparison Project - Haarsma et al. (2016) |
| **ISMIP6**: Ice Sheet Model Intercomparison Project for CMIP6 - Nowicki et al. (2016) |
| **LS3MIP**: Land Surface, Snow and Soil Moisture MIP - van den Hurk et al. (2016) |
| **LUMIP**: Land-Use Model Intercomparison Project - Lawrence et al. (2016) |
| **OMIP**: Ocean Model Inter-comparison Project - Griffies et al. (2016) |
| **PAMIP**: Polar Amplification Model Intercomparison Project - Smith et al. (2018) |
| **PMIP**: Paleoclimate Modeling Intercomparison Project - Kageyama et al. (2018) |
| **RFMIP**: Radiative Forcing Model Intercomparison Project - Pincus et al. (2016) |
| **ScenarioMIP**: Scenario Model Intercomparison Project - O'Neill et al. (2016) |
| **VolMIP**: Model Intercomparison Project on the climatic response to Volcanic forcing - Zanchettin et al. (2016) |

**Table 4.** The modelling CMIP6 experiments as introduced in Eyring et al. (2016). This list does not include CORDEX or the diagnostic MIPs, which are not currently included in the ES-DOC MIP documentation.

Table 4 provides a summary of most of the CMIP6 "endorsed" MIPs as of December 2018, with the DECK incorporated in CMIP as discussed above. This table was autogenerated from the ES-DOC experiment repository (see section 6 for the code). It does not include the MIPs that have not originated any of the CMIP6 experiments. There are three of these, which focus on and use CMIP6 output for various purposes: the Coordinated Regional Climate Downscaling Experiment (CORDEX, Gutowski Jr. et al. 2016 or the three diagnostic MIPs - DYnVarMIP (Dynamics and Variability MIP, Gerber and Manzini 2016), SIMIP (Sea Ice MIP, Notz et al. 2016) and VIACSAB (Vulnerability, Impacts, Adaptation and Climate Services Advisory Board, Ruane et al. 2016), as these are not yet included in ES-DOC. There are of course many other "non endorsed" MIPs such as ISA-MIP (the Interactive Stratospheric Aerosol MIP Timmreck et al. 2018) which could also be documented with the ES-DOC system at some future time.

## 4.2 Common Experiments

Figure 5 shows the sharing of experiments between MIPs. The importance of *piControl*, *historical*, *AMIP*, key scenario experiments (*ssp245* and *ssp585*), and the idealised experiments (*1pctCO2* and *abrupt-4xCO2*) is clear. These seven experiments form part of the protocol for many of the CMIP6 MIPs (Fig. 7). The scope of the *historical* and *piControl* experiments is demonstrated by their connections to MIPs on the far edges of the plot in all directions.

There are other shared experiments too, which bring MIPs together around shared scientific goals: *land-hist* jointly defined and shared by LUMIP and L3SMIP; *past1000* defined by PMIP forms part of VolMIP; *piClim-control* defined by RFMIP forms part of AerChemMIP; and *dcppC-forecast-addPinatubo* defined by DCPP forms part of VolMIP. By contrast, OMIP stands alone, sharing no experiments with other MIPs.

## 4.3 Common Forcing

Experiments share forcing constraints, just as MIPs share experiments. Figure 6 shows the interdependence of the DAMIP experiments on common forcing constraints. Experiments are grouped near each other when they share forcing constraints. The dense network shown reflects the similarity of experiments within DAMIP, and arises from a common design pattern/protocol in numerical experiment construction: a new experiment is a variation on a previous experiment with one (or a few) forcing changes. It is of course this "perturbation experiment" pattern which provides much of the strength of simulation in exposing causes and effects in the real world.

Unique modifications appear in Fig. 6 as forcing constraint nodes that are only connected to one or two experiments which is also why the alternative forcing experiments *hist-all-nat2* and *hist-all-aer2* are placed further from the main body of the DAMIP network — they share fewer forcing constraints with the other experiments. However, they themselves are similar to each other as between them they share a number of unique forcing constraints.

The importance of the perturbation experiment pattern is further emphasised in DAMIP by noting that the three external experiments (piControl, historical and ssp245) account for 62 percent of the DAMIP forcing constraints; five of the DAMIP experiments can be completely described by forcing constraints associated with these external experiments — being different

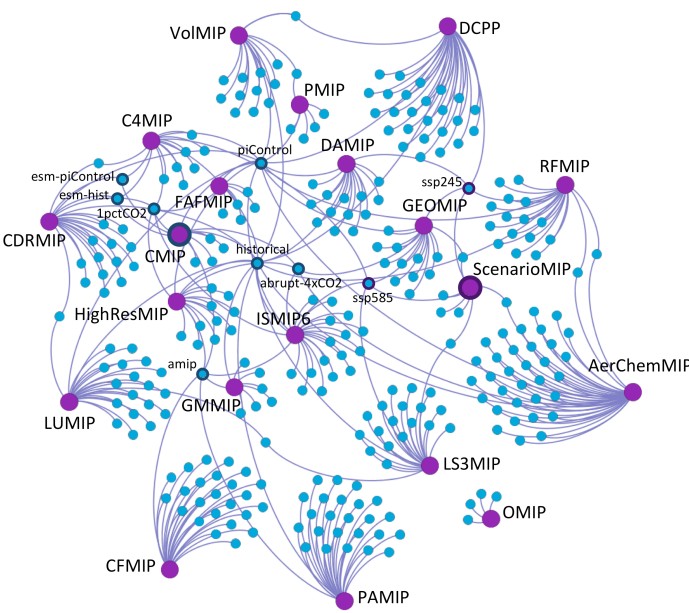

**Figure 5.** *CMIP6 MIPs and experiments.* Individual MIPs are represented by large purple dots. Lines connect each MIP to the experiments that are related to it, which are shown as smaller blue dots. Some widely used experiments are labelled, such as the piControl, historical, amip, ssp245 and ssp585, which are used by numerous MIPs within CMIP6.

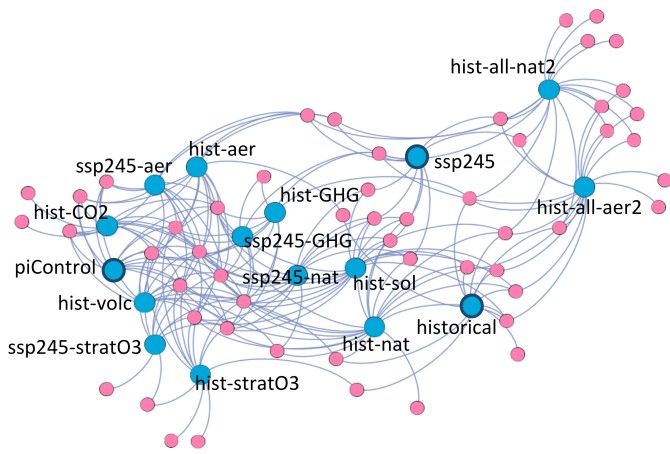

**Figure 6.** *DAMIP experiments and forcing constraints.* Individual experiments are represented by large blue dots. Lines connect each experiment to related forcing constraints, represented by pink dots. An example of a forcing constraint might be a constraint on atmospheric composition such as a requirement for a particular concentration of atmospheric carbon dioxide. In this figure three experiments are shown with dark blue borders (piControl, historical and ssp245) these are experiments that are required by DAMIP but are not defined by DAMIP. The forcing constraints for these three "external" experiments are used extensively by the DAMIP experiment suite.

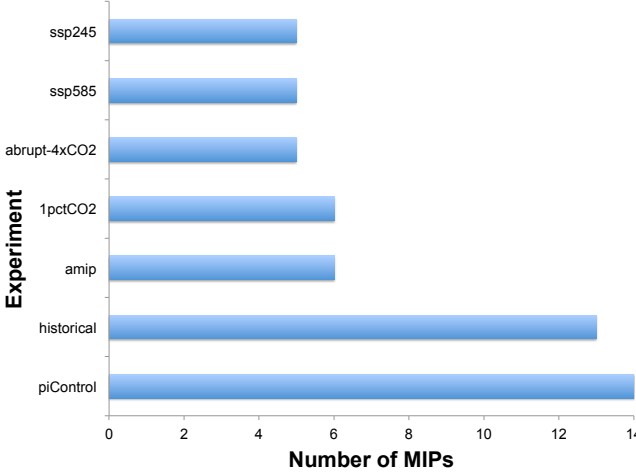

**Figure 7.** The most-used CMIP6 experiments in terms of the number of Model Intercomparison Projects (MIPs) to which they contribute.

assemblies of the same "forcing building blocks". The key role of these building blocks is exposed by placing the DAMIP experiments into sets according to which of those external experiments is used for forcing constraints (Fig. 8).

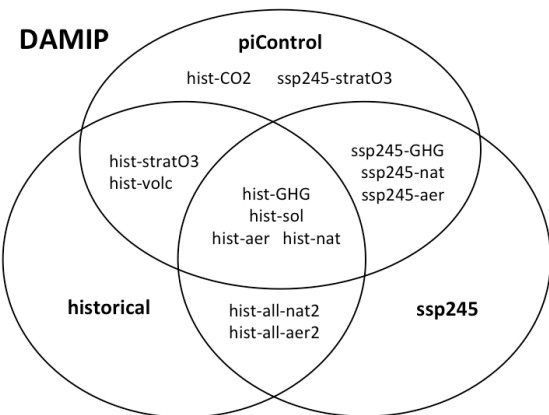

**Figure 8.** A view of *DAMIP* with experiments placed in sets according to the forcing constraints they share with the external experiments: piControl, historical and ssp245.

This framing of shared forcing constraints exposes some apparent anomalies. Why, for example, is *hist-CO2* not in the "historical" set? The reasons for these apparent anomalies expose the framing of the experiments. In the *historical* experiment, greenhouse gas forcing is a single constraint which includes CO2 and other well mixed greenhouse gases. By contrast, *hist-CO2* varies only CO2, with the other well mixed greenhouse gases constrained to pre-industrial levels (and hence uses the *piControl* forcing constraints for those, with it's own CO2 forcing constraint).

It would have been possible to avoid this sort of anomaly by constructing finer constraints in the case of *historical*, but this would have been at the cost of simplicity of understanding (and greater multiplicity in reporting as discussed below). There is a necessary balance between clear guidance on experiment requirements, and re-use of such constraints to expose relationships between experiments.

### 4.4 Forcing Constraint Conformance

One of the goals of the constraint formalism is to minimise the burden on modelling groups. Minimising the burden of executing the CMIP6 experiments and the burden of documenting how the experiments were carried out (that is, populating the concrete part of the experiment definition, using the language of Mattoso et al. 2010, as discussed in section 2.4). By clearly identifying commonalities between experiments, modelling groups can implement constraints once, and reuse both the implementation and documentation across experiments.

Constraint "conformance" documentation is intended to provide clear targets for interpreting the differences between simulations carried out with different models. Given that differing constraints often define differing experiments, understanding why models give different results can be aided by understanding differences in constraint implementation (in those cases where there is implementation flexibility). Section 3.3 discussed some aspects of this from a constraint definition perspective.

One can then ask, how much re-use of constraints is possible? Figure 9, shows that a few forcing constraints are re-used widely across CMIP6. These are the forcing constraints associated with the DECK and historical experiments and the prominent scenario experiments from ScenarioMIP which are used by numerous MIPs (Fig. 5). It is with these forcing constraints that many deep connections between MIPs are made. From a practical perspective the wide application of these forcing constraints allows for considerable streamlining of the documentation burden on CMIP6 modelling groups. Beyond this core we see a smaller group of forcing constraints that are used by a few MIPs. For the most part these are forcing constraints associated with the less prominent scenario experiments from ScenarioMIP. The remainder of the forcing constraints are specific to just one MIP, and of these, 265 are only used once by a single experiment. Although this last group of forcing constraints is large in number, many groups will only make use of them if they happen to run the specific experiments to which they pertain.

### 4.5 Temporal Constraints

History suggests there has been — and continues to be — divergent understanding of instructions for the expected duration of simulations (temporal constraints), often manifest by delivering "off by one" differences in the number of years of simulation. Such errors hamper statistical inter-comparison between simulations, and can result in unnecessary effort (often expensive in human and computer time). The CMIP6 experiments have not been immune from this issue. Temporal constraints in the CMIP6 controlled vocabulary are defined in terms of a start year and a minimum length of simulation expressed in years. However, the publications by the CMIP6-Endorsed MIPs often also include an end year which can be inconsistent with the minimum simulation length as described by the CMIP6-CV. The divergence in understanding generally occurs in the interpretation of the dates implied by a given start year and end year, specifically whether they refer to the beginning of January or the end of December.

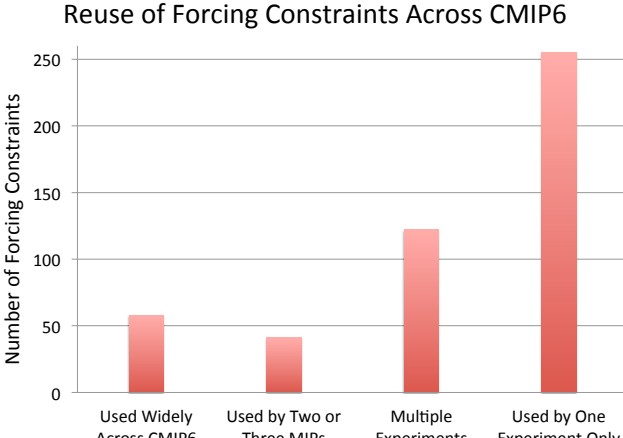

**Figure 9.** *Distribution of forcing constraint reuse across CMIP6.* Forcing constraints are categorised in terms of how widely they are used. Widely used forcing constraints are used by experiments in four or more MIPs.

A significant effort has been made by ES-DOC to identify these discrepancies and instigate their correction. ES-DOC temporal constraints unambiguously specify a start date, end date and length for simulations and are a mandatory part of the ES-DOC experiment documentation. Despite these steps, there are still many cases where the MIPs of CMIP6 might have coordinated yet further and used the same temporal constraints for different experiments with essentially the same temporal requirements, such as those that begin in the present day and run to the end of the 21st century. These differences provide scope for further rationalisation in future experiments and/or CMIP phases, leading to further simplification in analysis and savings in computer time.

## 5 Summary and Further Work

The need for structured documentation constrained by controlled descriptive terminology is not always well understood by all parties involved in creating content. While structured scientific metadata has an important role in science communication, it exacts a cost in time, energy and attention. This cost causes friction in the scientific process even though it can provide the information necessary for investigators to reach a common understanding across barriers arising from distance in space, time, institutional location, or disciplinary background. The balance between this "metadata friction" and the potential benefit in ameliorating the "scientific friction" barriers is difficult to achieve (Edwards et al., 2011). Solutions need to be iterative and achieve a balance between ease of information collection and structures which support handling information at scale and being able to support multi-disciplinary cross-walks in meaning. However, with the right information in place, it is possible to provide traceable, documented answers to questions about experiment protocols that could otherwise elicit different answers

from different individuals over time. Overall this can result in a reduction in the support load on all parties from those who designed the experiment to those who manage the simulation data.

In this paper we have introduced the ES-DOC structures for experimental design and shown their application in CMIP6. We have introduced a formal taxonomy for experimental definition based around collections of climate modelling projects (MIPs), experiments, and numerical requirements and, in particular, constraints of one form or another. These provide structure for the formal definition of the experiment goals, design and method. The conformance, model, and simulation definitions (to be fully defined elsewhere) will provide the concrete expression of how the experiments were executed.

The construction of ES-DOC descriptions of CMIP6 experiments has been carried out mostly by the ES-DOC team, using published material, but often as part of the iterative discussions which specified the CMIP6 MIPs. These iterative discussions, led by the MIP teams, with coordination provided at various stages by the CMIP panel and PCMDI, have improved on previous MIP exercises, albeit with a larger increase in process and still with opportunities for imprecision, duplication of design effort, and unnecessary requirements on participants. The ES-DOC experiment definitions provided another route to internal review of the design and aided in identifying and removing some of the imprecision, duplication of effort, and simulation requirements. However, there is still scope for improving the design phase.

Earlier involvement of formal documentation, would have facilitated more interaction between the MIP design teams by requiring more information to be shared earlier. Doing so in the future might allow more common design patterns, and perhaps more experiment and simulation re-use between MIPs, reducing the burden on carrying out the simulations and on storing the results. This potential gain would need to be evaluated and tensioned against the potential process burden, but it can be seen that the ES-DOC experiment/requirement/constraint definitions are relatively lightweight, yet communicate significant precision of objective and method. Early involvement of formal documentation is important for building a culture of engagement. Our experience with the CMIP6 MIPs indicates that the process of providing detailed information about experiments was perceived in a positive way by groups when the intervention occurred early in the experiment life cycle. These groups also had a sense of ownership of their content. In contrast, groups who engaged later in the experiment life cycle were more likely to perceive the documentation effort as yet another burden.

Sharing of experiments and constraints is clearly common within CMIP6, but there remain opportunities for improvement in this regard. Section 2.4 outlines a set of important relationships between the MIPs, and MIP dependency on key experiments — most of which are in the CMIP (and DECK) sub-project. Such sharing introduces extra problems of governance: who owns the shared experiment definition? In the case of the dependencies on the DECK, this is clear, it is the CMIP panel, but for other cases it is not so clear. For example, both LS3MIP and LUMIP needed a historical land experiment, and it was obvious it should be shared. In this case (and hopefully most cases) the solution was amicable, resulting in the description:

Start year either 1850 or 1700 depending on standard practice for particular model. This experiment is shared with the LS3MIP, note that LS3MIP expects the start year to be 1850.

Although clear, this is not really ideal for downstream users (either those who may run the simulations in the wrong order, or those analysts doing intercomparison). If sharing is to be enhanced in future CMIP exercises, then the early identification of synergies (and the resolution of any inconsistencies and related governance issues) will be necessary.

The sharing and visualisation of constraint dependencies (section 4.3) provides a route to both efficient execution and better understanding of experimental structure. In the case of DAMIP there is clear value to the interpretation of the MIP goals in terms of the forcing constraints, and this sort of analysis could both be extended to other MIPs and used during future design phases. While there is a trade-off between granularity of forcing and the burden of conformance documentation, with CMIP6 this trade-off was never explicitly considered. In the future it is possible that such consideration may in fact improve experimental design. We believe it will be easier for both the MIP designers and participants to be confident that they have requested, understood, and/or executed experiments that will meet their scientific objectives.

ES-DOC remains a work in progress. It is fair to say that there was not wide community acceptance of the burden of documentation for CMIP5, but this was in part because of the tooling available then. With the advent of CMIP6, the tooling is much enhanced, and available much earlier in the cycle, but both the underlying semantic structure and tooling can and will be improved. There is clearly opportunity of convergence between the Data Request and ES-DOC and there will undoubtedly be much community feedback to take on board!

ES-DOC is not intended to apply only to CMIP exercises. We believe the preciseness and self-consistency ES-DOC imposes on experiment design documentation should be of use even when only one or a few models generate related simulations. One such target will be the sharing of national resources to deliver extraordinarily large and expensive simulations (in time, resource, and energy) where individuals and small communities could not justify the expense without sharing goals and outputs. Realising such sharing opportunities is often impaired by insufficient communication and documentation. We believe the ES-DOC methodology can go some way towards capitalising on these opportunities and will become essential as we contemplate using significant portions of future exascale machines.

## 6 Code Availability

All the underlying ES-DOC code is publicly available at https://github.com/es-doc. The full CMIP6 documentation is available online at https://search.es-doc.org/. The ES-DOC documentation of the CMIP6 experiments can be found in the ES-DOC GitHub repository at https://github.com/ES-DOC/esdoc-docs/blob/master/cmip6/experiments/spreadsheet/experiments.xlsx. The code to extract and produce the ES-DOC tables in this paper is available online at https://github.com/bnlawrence/esdoc4scientists (Lawrence, 2019). Figures 5 and 6 were produced using content (in the form of triples) generated from ES-DOC and imported into gephi (https://gephi.org/) with manual annotations.

## Appendix A: Examples

To improve readability, a number of examples are provided in this appendix, rather than where first referenced in the main text.

All these tables are produced by a python script. The ES-DOC pyesdoc[5] library is used to obtain the documents and instantiate them as python objects with access to CIM attributes via instance attributes with CIM property names. These can then be used to populate HTML tables described using jinja2[6] templates which are then converted to PDF for inclusion in the document using the weasyprint [7] package. This methodology is more fully described in the code (Lawrence, 2019).

| **abrupt-4xCO2 (CMIP, DECK, AerChemMIP, GeoMIP, HighResMIP, ISMIP6)** | |
|---|---|
| *Abrupt quadrupling of the atmospheric concentration of carbon dioxide* | |
| **Description:** Impose an instantaneous quadrupling of the concentration of atmospheric carbon dioxide from the global annual mean 1850 value, then hold fixed. | |
| **Rationale:** To evaluate the effective climate sensitivity of the model (EfCS) and to diagnose the strength of various feedbacks. To characterise the radiative forcing that arises from an increase in atmospheric CO2 as well as changes that arise indirectly due to the warming. One can use the effective climate sensitivity to estimate the equilibrium climate sensitivity (EqCS). | |
| **Requirements** | |
| **Pre-Industrial Solar Particle Forcing**: Pre-Industrial solar particle forcing (1850-1873 mean). For models with interactive stratospheric chemistry. Proton forcing: HOx and NOx production by solar protons. Electron forcing: Kp- or Ap-index to describe ionisation from electron precipitation in the lower thermosphere and upper mesosphere. Cosmic ray forcing: ion-pair production by galactic cosmic rays. CMIP6 models that do not have interactive chemistry should prescribe the CMIP6 recommended ozone forcing data set. *Additional Requirements:*<br>• Pre-Industrial Proton Forcing<br>• Pre-Industrial Electron Forcing<br>• Pre-Industrial Cosmic Ray Forcing<br>• Pre-industrial stratospheric Ozone concentrations as a substitue for solar particle forcing for models without interactive chemistry | **Pre-Industrial Forcing Excluding CO2 and Solar**: Pre-Industrial forcing excluding carbon dioxide (CO2) and solar forcing. *Additional Requirements:*<br>• Pre-Industrial Well Mixed Greenhouse Gas (WMGHG) Concentrations excluding CO2<br>• Pre-Industrial Aerosols<br>• Pre-Industrial Aerosol Precursors<br>• Pre-Industrial Ozone Concentrations<br>• Pre-Industrial Stratospheric Water Vapour Concentrations<br>• Pre-Industrial Stratospheric Aerosol<br>• Pre-Industrial Land Use |
| **Pre-Industrial Solar Irradiance Forcing**: Pre-Industrial solar forcing. The standard solar forcing dataset recommended for usage is the solar reference scenario dataset which includes pre-industrial solar forcing (1850-1873 mean). Includes total solar irradiance, F10.7 cm solar radio flux, and spectral solar irradiance for 10-100000 nm range. | **Abrupt 4xCO2 Increase**: Impose an instantaneous quadrupling of atmospheric carbon dioxide concentration, then hold fixed. |
| **PreIndustrialInitialisation**: Initialisation from a January in the pre-industrial control simulation. | **AOGCM Configuration**: Use a coupled Atmosphere-Ocean general circulation model |
| **SingleMember**: One ensemble member | **150yrs**: Run for 150 years. |

**Table A1.** The abrupt 4XCO2 experiment is integral to a number of MIPs. (Not all properties are shown, see http://documentation.es-doc.org/cmip6/experiments/abrupt-4xCO2 for more details.)

---

[5]https://pypi.org/project/pyesdoc/

[6]http://jinja.pocoo.org/

[7]https://weasyprint.org/

| land-noFire (LUMIP) |
|---|
| *historical land-only with no human fire land management* |
| **Description:** Land surface model simulation. Same as land-hist except with fire management maintained at 1850 levels. Start year either 1850 or 1700 depending on standard practice for particular model. |
| **Rationale:** To assess the relative impact of land cover and incremental land management change on fluxes of water, energy, and carbon in combination with other LUMIP land experiments. |
| **Requirements** |
| **1700-2014 315yrs**: Historical, from 1700 to 2014. |
| **1850-2014 165yrs**: Historical, pre-Industrial to present |
| **Historical GSWP3 Meteorological Forcing**: Apply Global Soil Wetness Project phase three (GSWP3) forcing data for offline land surface models running the LS3MIP historical simulation land-hist is provided by the LS3MIP. |
| **Historical Land Use**: Apply the global gridded land-use forcing datasets to link historical land-use data and future projections. This new generation of "land use harmonization" (LUH2) builds upon past work from CMIP5, and includes updated inputs, higher spatial resolution, more detailed land-use transitions, and the addition of important agricultural management layers. |
| **Historical land surface forcings except fire management**: Apply all transient historical forcings that are relevant for the land surface model except for fire management. |
| **1850 Fire Management**: Maintain 1850 levels of fire management (anthropogenic ignition and suppression of fire). If ignitions are based on population density, maintain constant population density. |
| **SingleMember**: One ensemble member |
| **LSM Configuration**: Offline land surface model |
| **All Land Management Active**: All applicable land management active in the land surface model configuration. |

**Table A2.** This is an experiment that has an anti-forcing "Historical Land Surface Forcings Except Fire Management" (note also two temporal constraint options "Start year either 1850 or 1700 depending on standard practice for particular model."). See https://documentation.es-doc.org/cmip6/experiments/land-NoFire for more information.

| **G7SST1-cirrus (GeoMIP)** |
|---|
| *SSTs from year 2020 of SSP5-8.5; forcings and other prescribed conditions from year 2020 of SSP5-8.5 + cirrus thinning* |
| **Description:** Time slice at year 2020 of GeoMIP G7cirrus. Run for 10 years. |
| **Rationale:** To assess radiative forcing of G7cirrus at the beginning of the simulation (2020). |
| **Requirements** |
| **2020-2029 10yrs**: Timeslice, begin in 2020 and run for 10 years. |
| **Increase Cirrus Sedimentation Velocity**: Add a local variable that replaces (in all locations where temperature is colder than 235K) the ice mass mixing ratio in the calculation of the sedimentation velocity with a value that is eight times the original ice mass mixing ratio. Cirrus seeding to begin in 2020 and continue through to the year 2100. |
| **SSP5-85 SST 2020**: Sea surface temperature climatology calculated from the ScenarioMIP SSP5-85 experiment for the year 2020. |
| **SSP5-85 SIC 2020**: Sea ice concentration climatology calculated from the ScenarioMIP SSP5-85 experiment for the year 2020. |
| **RCP85 Forcing**: Impose RCP8.5 forcing.<br>*Additional Requirements:*<br>• Representative Concentration Pathway 8.5 Well Mixed Greenhouse Gases<br>• Representative Concentration Pathway 8.5 Short Lived Gas Species<br>• Representative Concentration Pathway 8.5 Aerosols<br>• Representative Concentration Pathway 8.5 Aerosol Precursors<br>• Representative Concentration Pathway 8.5 Land Use for Shared Socioeconomic Pathway 5 |
| **SingleMember**: One ensemble member |
| **SSP5-85Initialisation2020**: Initialisation is from the beginning of year 2020 of the SSP5-8.5 experiment. |
| **AGCM Configuration**: An Atmosphere only general circulation model configuration. |

**Table A3.** The "Increase Cirrus Sedimentation Velocity" forcing constraint is very precise about the change to be made to the "Add a local variable that replaces (in all locations where temperature is colder than 235K) the ice mass mixing ratio in the calculation of the sedimentation velocity with a value that is eight times the original ice mass mixing ratio". See https://documentation.es-doc.org/cmip6/experiments/g7sst1-cirrus for more information.

| G6sulfur (GeoMIP) |
|---|
| *stratospheric sulfate aerosol injection to reduce net forcing from SSP585 to SSP245* |

**Description:** Injection of sulfate aerosol precursors in the equatorial stratosphere to reduce the radiative forcing of the ScenarioMIP high forcing scenario (SSP5-85) to match that of the ScenarioMIP medium forcing scenario (SSP2-45). Geoengineering will be simulated over years 2020 to 2100.

**Rationale:** To evaluate a climate in which geoengineering is used to only partially offset climate change in order to reduce the burden of adaptation. Assess the climate effects and inter-model variations of a limited amount of geoengineering as part of a portfolio of responses to climate change. Results to be compared with G6solar to determine differences between sulfate aerosol effects and solar irradiance effects.

| Requirements | |
|---|---|
| **Internal Stratospheric Aerosol Precursors RCP85 to RCP45**: Injection of stratospheric sulfate aerosol precursors to reduce the radiative forcing of ScenarioMIP high forcing scenario (SSP5-85) to match that of the ScenarioMIP medium forcing scenario (SSP2-45). Modelling groups that have an internal sulfate aerosol treatment should calibrate the radiative response to sulfate aerosols individually so that the results will be internally consistent (a procedure that will be more difficult for models that have a complex microphysical treatment of aerosols). Potential methods include a double radiation call, once with and once without the stratospheric aerosols, and also the use feedback methods. Simulations to be conducted as if the aerosols or aerosol precursors are emitted in a line from 10°S to 10°N along a single longitude band (0°). The injected aerosols or aerosol precursors should be evenly spread across model layers between 18 and 20 km. Note that sedimentation processes and self-lofting due to heating are likely to result in the aerosols being distributed between 16-25 km in altitude. | **External Stratospheric Aerosol Precursors RCP85 to RCP45**: Injection of stratospheric sulfate aerosol precursors to reduce the radiative forcing of ScenarioMIP high forcing scenario (SSP5-85) to match that of the ScenarioMIP medium forcing scenario (SSP2-45). For modelling groups that have no dynamical treatment of sulfate aerosols, GeoMIP will provide a data set of aerosol optical depth, as well as ozone fields that are consistent with this aerosol distribution. Note that the amount of sulfate injection needed for a given model to achieve the goals of this experiment may vary, so modelling groups should scale the aerosol and ozone perturbation fields as necessary. Simulations to be conducted as if the aerosols or aerosol precursors are emitted in a line from 10°S to 10°N along a single longitude band (0°). The injected aerosols or aerosol precursors should be evenly spread across model layers between 18 and 20 km. Note that sedimentation processes and self-lofting due to heating are likely to result in the aerosols being distributed between 16-25 km in altitude. |
| **RCP85 Forcing**: Impose RCP8.5 forcing.<br>*Additional Requirements:*<br>• Representative Concentration Pathway 8.5 Well Mixed Greenhouse Gases<br>• Representative Concentration Pathway 8.5 Short Lived Gas Species<br>• Representative Concentration Pathway 8.5 Aerosols<br>• Representative Concentration Pathway 8.5 Aerosol Precursors<br>• Representative Concentration Pathway 8.5 Land Use for Shared Socioeconomic Pathway 5 | **SSP5-85Initialisation2020**: Initialisation is from the beginning of year 2020 of the SSP5-8.5 experiment. |
| **AOGCM Configuration**: Use a coupled Atmosphere-Ocean general circulation model | **2020-2100 81yrs**: Scenario, from 2020 to the end of the 21st century |
| **SingleMember**: One ensemble member | |

**Table A4.** GeoMIP is clear about what the forcing should achieve (reduction in radiative forcing from rcp8.5 to rcp4.5) but leave it open to the modelling groups to choose a method that best suits their aerosol scheme. See https://documentation.es-doc.org/cmip6/experiments/g6sulfur for more information.

*Author contributions.* CP represented ES-DOC in discussions with CMIP6 experiment designers, collecting information and influencing design. MJ was responsible for the data request. KT led the PCMDI involvement in experiment coordination. EG and BL led various aspects of ES-DOC at different times. BL and CP wrote the bulk of this paper, with contributions from the other authors.

*Competing interests.* The authors declare that they have no conflict of interest.

5  *Acknowledgements.* Clearly the CMIP6 design really depends on the many scientists involved in designing and specifying the experiments under the purview of the CMIP6 panel. The use of ES-DOC to describe experiments depends heavily on the tool chain, much of which was designed and implemented by Mark Morgan under the direction of Sébastien Denvil (CNRS/IPSL). Paul Durack was instrumental in the support for CMIP6 vocabularies at PCMDI. Most of the ES-DOC work described here has been funded by national capability contributions to NCAS from the UK Natural Environment Research Council (NERC) and by the European Commission under FW7 grant agreement
10  312979 for IS-ENES2. The writing of this paper was part funded by the European Commission via H2020 grant agreement No 824084 for IS-ENES3. Work by Karl E. Taylor was performed under the auspices of the US Department of Energy (USDOE) by Lawrence Livermore National Laboratory under contract DE-AC52-07NA27344 with support from the Regional and Global Modeling Analysis Program of the USDOE's Office of Science.

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
