# Peer review of "Documenting numerical experiments in support of CMIP6"

_Geoscientific Model Development, 2019_

## Referee Comment (RC1) · Ron Stouffer (Referee) · 7 Jul 2019

General Comments

My biggest general comment is that the title is very misleading. I could see where some scientists involved in designing the CMIP6 experiments/simulations would be surprised that they did not do what is claimed in the title. The title needs changed. My recommendation is "A system for documenting models and experiments in CMIP6". While I agree the authors had a big positive impact on the simulation implementation and in some cases design in CMIP6. This by identifying problems with design, variable list and etc., they should not take credit for the whole thing...especially the science aspects as the current title suggests. The authors acknowledge this fact in the paper –

see page 3, lines 30 – 31 as one example.

My last detailed point on effective and equilibrium climate sensitivity is also very important. See below.

Finally, I have lots of wording suggestions for the authors to consider. Most are trying to improve the clarity of the discussion.

I recommend that the paper be published with major revisions. I would like to see the paper again before it is accepted. I think all of my changes are relatively easy to address. Normally a review of this kind would be minor revisions. However, I feel that the current title or the current ECS discussion is important enough to change my recommendation to major revisions.

Detailed Comments

1. Page 1, Title – Important! - See above.

2. Page 1, Lines 2-3 – editorial - may communicate primarily – Change to "typically communicate". Reads better and is closer to what happens.

3. Page 1, Lines 9-10 – expected methodology – I am not sure what this means. I think it means – paths to those goals.

4. Page 1, Line 10 – editorial – was intended – is intended.

5. Page 1, Line 19 – editorial clarity – Add "for MIPs" after "protocols".

6. Page 2, Figure 1 caption – The last 2 sentences in the figure need to be moved into the text. They are too important to leave in the caption.

7. Page 3 top – Needs a reference to figure 1a somewhere.

8. Page 3, line 5 – Add reference to Eyring et al. after "DECK".

9. Page 3, line 14 – conform as best they can – Change to "attempt to conform". Reads better.

10. Page 3, lines 30 – 31 – I note in passing that these lines make my point about the title being misleading.

11. Page 4, line 5 – heritage – Reference needed or define. It is not clear what is meant by the word – heritage.

12. Page 4, line 11 – editorial - shifted slightly in terms – Change to "shifted slightly from the past in terms". Clearer.

13. Page 5, line 1 – Thus the DCPP . . . - It is not clear to me why this follows. Delete or make clear.

14. Page 5, figure 3 caption – Change "intercomparison projects" to "MIPs". Why introduce new nomenclature?

15. Page 7, line 2 – explicitly calling out the failure – More is needed here. Exactly what kind of information is missing? Give few examples.

16. Page 7, line 2 – Add "published" before "papers".

17. Page 7, line 4 – 6 – This paragraph hangs. Add more or delete. If kept, explain how the present structure improves on the past in some detail and/or examples.

18. Page 7, line 9 – Add "a controlled vocabulary (CV)" before "introduced in Mattoso".

19. Page 7, line 23 – I think adding "climate" before "experiments" makes things clear for the reader.

20. Page 8, line 19 – Add "climate modeling" between "major" and "centers".

21. Page 8, line 21 – driven – This is too strong. It implies the IPCC drives the process which is incorrect. The WCRP/WGCM/CMIP Panel drives the process with the IPCC timelines in view. Change "driven" to "associated with".

22. Page 8, lines 23 – 25 – Investigating differences in the models' response is missing from this list. It is the reason a 1% CO2 simulation was included and the main reason

for starting CMIP. The current list is very misleading.

23. Page 9, Table 3, Rationale for the DECK – Investigating the causes for differences in the models' response is missing again (see point 22 above). Again, this investigation was the main reason for starting CMIP. It continues to be important today. It is important that the rationale for these experiments be clear and accurate.

24. Page 9, line 6 – editorial – Change "project leaders" to "Panel". Clearer.

25. Page 10, lines 8 – 10 – The last sentence in this section hangs. More is needed. It needs to be clear that there are many MIPs ongoing outside of CMIP (more than 50, the last count I saw).

26. Page 10, line 12 – Add "experimental" before "design process". Clearer.

27. Page 14, lines 3 – 12 – forcing and temporal constraints need to be better defined. I think I understand what they are but am not sure. Some more examples would be helpful. Forcing constraints could be thought of as radiative forcing constraints, for example.

28. Page 14, line 7 – Assuming I understand things. . .add "i.e., length of simulation" after "temporal constraints".

29. Page 14, lines 10 – 12 – I do not understand the point here. Is the point that different MIPs and simulations use differing start and end dates. Or length of simulation? If so, what is the scientific problem? Is there one? Also, it seems that these details should be documented in ES-DOCs. I assume they are and if so, this then is an issue between the authors and the MIP leaders. . ..which makes no sense to me. I am lost.

30. Page 15, line 27 – What is a "triples"?

31. Page 23, 24, figures 5 and 6 – Both on my screen and in printed versions, the lines are very hard to see. The lines being hard to see means that the points make in the text are lost.

32. Page 26, rationale for the switch-on 4X simulation – The experiment does not define the equilibrium climate sensitivity (EqCS). It defines the effective climate sensitivity (EfCS). One can use the effective climate sensitivity to estimate the equilibrium climate sensitivity. See AR5 WG1 report for a discussion of this point. This is an important. Some in our community is using the two term interchangeably. This is causing problems. They are not the same thing. EfCO2 is a transient value (changes in time). EqCS is an equilibrium value, constant in time.

---

## Author Comment (AC1) · 1 Aug 2019

We thank Ron for his review.

In hindsight we agree that the title is misleading, we had not intended for it to imply that the authors had "designed CMIP6 experiments", but rather that the paper was about the process of design and documentation (which indeed it is).

We propose to retitle the paper something along the lines of "A process to facilitate the design and documentation of numerical experiments as applied to CMIP6". (The exact wording of our revised title may change when we have seen and responded to all the reviews.)

Most of the detailed comments require minor typographical improvements, which we

mostly agree with - we will detail the consequential changes with a collated response to all referees. There are a couple of substantive points which are actually do with the experiment descriptions themselves, which will be updated accordingly - we are grateful to Ron for the clarifications.

---

## Short Comment (SC1) · 7 Aug 2019

All, thanks for the chance to review this truly interesting paper.

I won't repeat Ron Stouffer's main comments, but I agree with them.

Broad comments: given that the experience of interacting with ES-DOC and its predecessors has often been tedious for the modeling groups, it's difficult to convey the power and significance of these tools for the MIPs. I think this paper does a good job of that, but overall the intro and sections 1 and 2 still sometimes read too much like a conversation with close colleagues in metadata management, rather than an exposition that would be widely understood by the larger community. Starting with Section 3 the paper opens up and flows better.

[Figure]

A few examples:

Section 2, p. 3 - lines 15-19 - what is the Data Request coordinator, and what sort of "input" does s/he provide? By the end of the article I understand the role that "additional documentation" could play in streamlining experimental design, but it's counterintuitive at this point.

P. 4 line 11- p 5 line 2: this paragraph seems to come out of nowhere, unattached to what precedes and follows it. Again, by the end of the article I can understand this, but not here, perhaps because it reads as if the Ấ ű is about the DCPP, when it's really about something more general.

Figure 3. Here we start to get the 0.1, 0.N, etc. notation, but it's not explained until Table 1 on the following page.

Section 2.3 - begins with a discussion of the ES-DOC controlled vocabulary, but this is a bit confusing since you have just spent 2 pages introducing your own, different vocabulary for talking about numerical experiments. Maybe simply saying "the ES-DOC controlled vocabulary" a few times early in this section would resolve the confusion.

Table 1 - probably your target audience will follow this, but I would have liked an explanation of what a "type" is.

P 7 lines 4-6 - this is aimed at your argument about the potential for ontologies to help with experimental design. That would work better if it flowed directly to the second half of the second sentence of the following paragraph, without the self-interruption about an updated ES-DOC ontology; that can go elsewhere.

P 7 lines 19-21 - this is an opportunity to say more about the issue of poorly constrained experiments in the simulation sciences, which is ubiquitous and in desperate need of clear answers. I think of your work, along with the MIPs themselves, as a crucial step in the direction of that answer; this could be brought more to the fore in your article. A brief discussion of the history of MIPs, and their epistemic importance, appears at pp.

349-352 of my book A Vast Machine: Computer Models, Climate Data, and the Politics of Global Warming (MIT Press, 2010).

Section 3.2 - p. 13 line 3: what I've really enjoyed hearing Bryan talk about is the human side of working with Metafor and similar tools (e.g. evidence of cutting and pasting in model descriptions as people become bored and inattentive during a tedious, detail-focused process). Here all of that is packed into one word - "infamous" (and a short paragraph on p 18, lines 21-25). I think it would be worth the space to discuss those experiences a bit more deeply, because they may affect modellers' ability and willingness to use the documentation for experimental design, as you're suggesting. I've attached a paper that might be of interest for your discussion of how metadata are actually used in science: Edwards, P. N., M. S. Mayernik, A. L. Batcheller, G. C. Bowker, and C. L. Borgman. 2011. "Science Friction: Data, Metadata, and Collaboration." Social Studies of Science 41 (5): 667–90.

Section 4.1 is really interesting.

Section 4.3 - last sentence on p. 16 — can you interpret this for us? What does it say about the potential for re-use of constraints, or perhaps about the particular experiments where constraints were not reusable?

Section 5, Summary - last sentence on p. 17 - is this earlier involvement realistic, given the pace of change and the hectic IPCC schedule? How would it start to gain a foothold in the community?

Detailed comments:

P 2 line 14: don't you mean "but also the experiments themselves" (that is, you mean ES-DOC helps not only with documenting experiments, but also with designing them)? Rephrasing the sentence would make this clearer. The logic of your "not only ...but also" here is hard to parse.

P 5 line 8 - should be "e.g. whether the model should be ..." to avoid embedding a

question in a statement.

P 8 line 7 - this might be better as "cannot be measured empirically," since simulations do generate (simulated) measurements.

P. 12 lines 14-15 - there's something wrong with the last half of this sentence - I can't follow what's been agreed by whom, or to what.

P 12, last 3 sentences: these could be much clearer.

P 13 line 14 - G6sulphur is misspelled

The whole paper needs a review of punctuation. There are commas where none should appear, commas that should be semicolons, and places where commas are needed.

Please also note the supplement to this comment:
https://www.geosci-model-dev-discuss.net/gmd-2019-98/gmd-2019-98-SC1-supplement.pdf

**Supplement:**

[supplement omitted: unrelated document]

---

## Short Comment (SC2) · 8 Aug 2019

**Comments from CMIP Panel and RFMIP members**

**General Comments**

The paper by Pascoe et al. on "Designing and Documenting Experiments in CMIP6" outlines a concept for the documentation of the experiment description in CMIP6. The authors describe ES-DOC as the place where the experiments are designed and documented, yet the original source of the experiment design and descriptions are the papers from the CMIP Panel (for the DECK and historical simulations) and the

[Figure]

CMIP6-Endorsed MIPs (for the CMIP6 experiments) in the GMD CMIP6 Special Issue and possibly new papers (version controlled) in case the experiment design changes or bugs are found. The paper should be changed to reflect that.

Until now a formal procedure has not been established for clarifying, modifying, and updating the CMIP6 experiment design as originally documented in the GMD paper descriptions. Furthermore, the narrative describing experiments in the GMD articles may not provide a comprehensive recipe for setting up and performing the experiments. The effort of the ES-DOC author team has been very helpful in identifying unclear details in experiments and between MIP projects, and they have a system which supports version control and traceability. However, ideally, the authority and responsibility to make changes lies with the MIP chairs, but this needs to be consistent with infrastructure requirements, and validated to minimize the introduction of ambiguity and/or further errors in specification. This requires software which can be used by the MIP teams to construct standardized descriptions which can enter the ES-DOC tool chain for presentation, validation and comparison, but as it stands there are no easy to use templates and software to do this.

Therefore, the following approach of experiment documentation has been agreed by the CMIP Panel and the WGCM Infrastructure Panel (WIP) in collaboration with the ES-DOC team (i.e. the authors of this paper):

- There will be a single authoritative source for the experiment design information that is version controlled with a DOI assigned (with content in a Github repository for each CMIP6-Endorsed MIP that holds this information).

- The MIP co-chairs will have the responsibility and authority to document the experiments and all changes in this central version controlled location.

- This documentation will conform in content with the ES-DOC specification and be formatted using a new template system to be constructed by a small working group, working under the joint auspices of the CMIP Panel and the WIP, which will use existing documentation as exemplars.

- When new documentation is produced or existing documentation updated, before any release and DOI assignment, there will be a review of the content by the WIP in consultation with the CMIP Panel in order to ensure that the information can smoothly and automatically propagate from the authorized source into the Data Request and the Controlled Vocabulary.

- The CMIP Panel will make it an Endorsement Criterion for MIPs to fill the experiment design in this single version controlled repository and to take responsibility and authority to maintain it up to date.

- The MIPs are also asked to keep the description of their experiment design up-to-date in the CMIP6 Special Issue at GMD. Corrigenda should only be used for mistakes in the description of the experimental protocol. Any changes in the experimental protocol itself, including bug fixes, require a new short paper of the Experiment Description paper type. New experiments or updates in the protocol also require a new paper. Such papers should highlight what has been changed compared to the previous version, give the reasoning behind the changes, with version numbering assigned so it is clearly traceable.

In CMIP6 a more distributed structure has been put in place (Eyring et al., doi:10.5194/gmd-9-1937-2016, 2016). As we move forward, we envisage making enhanced use of the distributed structure which includes the responsibility and authority of the experiment design description by the CMIP6-Endorsed MIPs supported by a central endeavor as proposed here. The value of ES-DOC lies in helping this process by defining templates and reviewing the experiment design and by bringing

information from different CMIP6-Endorsed MIPs together similar to what the authors present in Section 4. However, we need to work to avoid a data-user rather than data-provider imposed definition of the norms and standards on the community that the community cannot support.

The paper should be rewritten to reflect this agreed approach for CMIP. As it stands, the paper would be misleading to all those involved in CMIP.

**Specific Comments**

- The title needs changing. ES-DOC has not designed experiments in CMIP6. The experiments are designed by the CMIP Panel (DECK and historical simulation) and by the CMIP6-Endorsed MIPs. The paper is about the **taxonomy** or **nomenclature** of terms for more clearly defining experiments. Please change the title to reflect that.

- The manuscript would be stronger as literature, and the ES-DOC effort more compelling to the community, if the benefits to those undertaking the extra efforts (modelling centers, MIP chairs) were more clear. As it is, ES-DOC asks those producing the data to undertake quite a lot of effort. The benefits to those wanting to analyze the data are clear; the benefits to those producing the data are less so. Here is a chance to explain why they should also embrace this effort.

- The authors might also take the chance to explain why undertaking the very significant effort to comply with ES-DOC requests adds value to normal scientific communications. For example by explaining how ES-DOC provides a traceable, documented answer to questions about a simulation or protocol that might otherwise need to be answered multiple times and with potentially different responses

to email or other questions from analysts.

- In the manuscript and as implemented, the system would be more useful if it were more flexible. One wonders, for example, what value is brought by requiring information in ES-DOC for all the solar particle forcings when the experimental descriptions do not specify them. (This seems to be the root of one of the mistakes with respect to RFMIP experiment documentation in ES-DOC.)

- p2,l11: *"and ES-DOC use is now required for the documentation of CMIP6 simulations."* please reword to *"and the authors now recommended the use of ES-DOC for documentation of CMIP6 simulations."*

- p3,l10: The title of this section is misleading as the experiment are defined by the MIPs. Maybe the definition of the nomenclature?

- p3,l15 *"In the case of CMIP6, the iterative discussion includes input from the ES-DOC community aiming to get a formal experiment description, from the Data Request coordinator, and the CMIP6 central team at the Program for Climate Model Diagnosis and Intercomparison (PCMDI) responsible for cross-experimental common CMIP vocabularies. These extra activities result in additional documentation which can be used by those carrying out the actual experiment (figure 1b) in an attempt to minimize the burden of interpreting and carrying out many experiments."* Inaccurate description of how the experiment design in CMIP6 is established, namely by the CMIP6-Endorsed MIPs. Similarly, Figure 1 should be changed to reflect that the responsibility and authority of defining the CMIP6 experiments is with the CMIP6-Endorsed MIPs. The experiment design and the science comes down to the MIPs as was envisioned from the beginning of the planning for CMIP6 to make it a distributed effort where the MIPs and modeling groups have ownership on the design / models, respectively. ES-DOC should help this envisaged distributed process and could play the role of bringing documentation on different MIPs from the original authorized source together, but

should not request to be the authorized source as seems to be described in this paper.

- p10,l13 *"the CMIP6 team (both the CMIP panel1 and the PCMDI support group2)."* This is a wrong definition of the CMIP6 team, misleading wording, please rephrase. Similarly, please also avoid the use of *co-design between the CMIP Panel and PCMDI* in several places of the paper.

- p10,l11: The title of this section is misleading. The definition of the MIPs is a scientific undertaking by communities involved in specific science questions. Perhaps a better title would be 'Documentation of the experimental design process'.

- Section 6. We appreciate that the underlying code for ES-DOC is made publically available in a Github repository. However, when looking at the code the actual experiment description (i.e. the entries for the various experiments) seems not to be available. For traceability, it would be nice to have this all in an open Github repository as envisaged and described above.

**Comments on Figures and Tables**

- Figures 1 and 2 need to be changed as the workflow is not accurately representing the workflow in CMIP6.

- Table 4 on the CMIP6-Endorsed MIPs should be removed.

We encourage the authors to take our comments and the general longer-term visions for CMIP experiment description into account when rewriting the paper. If not considered, we recommend that the paper is either rejected, or another example than CMIP6 is taken and the paper gets removed from the GMD CMIP6 Special Issue.

Best regards,

*Veronika Eyring, Greg Flato, Jean-Francois Lamarque, Gerald Meehl, Robert Pincus, Cath Senior and Bjorn Stevens (Members of the CMIP Panel and RFMIP)*
* * *

---

## Author Comment (AC2) · 4 Sep 2019

We thank Paul for his set of detailed comments on our paper. We think addressing these will considerably strengthen the paper. We will itemise how we have addressed them in our collated response to all reviews.
* * *

---

## Author Comment (AC3) · 4 Sep 2019

We thank Veronika and her co-authors for their review of our paper.

It is clear that our working title has misled this group as well (as seen in RC1 and our response). The paper does not outline a concept for the documentation, it outlines the \*existing methodology\*, and the outcomes of the \*data repository\* documentation of CMIP6 which has been produced \*thus far\*. It is most certainly not intended as a formal manifest for what should be done in the future, although it does include some recommendations which reflect the opinions of the authors. We will ensure that along with the title change, we will re-word where possible to make this distinction clear.

There are two sorts of comment within this review: those which pertain to the technical

content, and those which pertain to the governance of the CMIP process. We will itemise our response to the former (technical content) in our consolidated response to all the reviews and comments. With respect to the governance, we *recommend that they should be dealt with elsewhere i.e. not in academic literature*; such issues are itemised here:

1) We did not intend to "describe ES-DOC as the place where experiments are designed and documented", we had intended to describe the ES-DOC role in harmonising experimental design, and to describe the documentation itself. We will endeavour to make this more clear in the revised version. The issue of which repository is considered to host the authoritative documentation is a governance issue, which we will not address here, but we will simply note that formal metadata (not published papers) describing the data are a requirement for some of the ESGF data repository funding, so ES-DOC metadata is a necessary *component* of CMIP support.

2) The authors of this short comment have confused a plan for the future (regarding the collection, *harmonisation*, and recording of documentation), discussed offline with *some* of the authors of this paper, with this paper, which as outlined above, is a description of what has been done for CMIP6, not what might be done for future CMIP phases. We do not intend to rewrite this paper to turn it into a position paper, the work and methodology done is what it is.

3) The CMIP panel may wish to avoid a data-user rather than data-provider imposed definition of norms and standards, but that is not currently possible given much of the funding which supports the CMIP process. How that is resolved is clearly a governance issue beyond the scope of this technical paper.

4) We had not submitted this paper to the CMIP6 special issue, and do not see the need for it to appear there, but it is a paper about the technology and processes which have supported CMIP6 *thus_far*, and so we do not think it should be altered to use a different example.

---

## Author Comment (AC4) · 19 Sep 2019

1. All those who responded want a title change. Our revised/proposed title is "A process to facilitate the design and documentation of numerical experiments as applied in support of CMIP6." 2. We will try and remove any implications that the ES-DOC team designed the CMIP6 experiments, that was clearly not what happened, and not what the text intended to say. We will however, continue to make it clear where the ES-DOC team contributed to the iterative development of experiments by requesting and adding clarity, and harmonising approaches across MIPS through the provision of a common format and vocabulary for the technical descriptions of the experiments. 3. There were corrections requested to the scientific content of tables 3 and 5, we have made those corrections in the source ES-DOC documents and will re-generate the tables for the

paper. 4. We intend to re-order and rewrite the front material to more clearly address the benefits to all parties of the ES-DOC approach. In doing so, we will a. review figure 1, and either modify it, or rewrite the caption and accompanying text to make more clear who did what, and clarify responsibilities, and b. address the comment that the paper is addressing metadata colleagues not the wider community. 5. We note that the experiment descriptions themselves are also available in a GitHub repository, and the code provided actually retrieves the JSON versions directly - however, the text doesn't explicitly say that. The revised version will include clear signposting as to how to find the source content. 6. SC2 requested that we remove table 4, but provided no rationale. We included this table as an example of how the ES-DOC repository can be used to extract a complete tabular description of CMIP6 from the documentation within (and the code provided shows how this was done). It is not clear why this use-case should be excluded, and in fact we think it shows how the ES-DOC documentation could be used in future CMIP (or similar) activities to keep a dynamic and up-to-date description of agreed experiments/MIPs. Accordingly, we will leave it in unless the Editor suggests differently. 7. There are many other minor corrections and useful suggestions for improvement throughout the comments. We intend to address nearly all these as suggested, and will itemise how we did that alongside the revised version of the manuscript.
* * *

---

## Author Response (AR1)

**Detailed point-by-point response from the authors of gmd-2019-98**

In responding to the reviewers comments we have made substantial changes to our manuscript, in particular the introductory sections have had a thorough revision. Rather than leading with examples from CMIP6, we now begin with a more general discussion about the challenges of running simulations in the Earth system sciences and set a context for our work in terms of the workflow of running those simulations. We believe that this context setting gives the paper a broader appeal and goes some way towards clarifying why the documentation effort is useful. We believe we have made clear the important distinction between the design of experiments by the CMIP6-Endorsed MIPs and the work of interpreting and clarifying them which has been done by the infrastructure teams working to support CMIP6.

| Reviewer | Comments from referees/ public | Author's response / changes in manuscript | Author's changes in the manuscript |
|---|---|---|---|
| RC1, SC2 | Rephrase title of paper | We have changed the title to "Documenting numerical experiments in support of CMIP6" | Documenting numerical experiments in support of CMIP6 |
| SC1 | The whole paper needs a review of punctuation. There are commas where none should appear, commas that should be semicolons, and places where commas are needed. | We have reviewed the text and corrected the punctuation errors and inconsistencies that we found. | |
| SC2 | The manuscript would be stronger as literature, and the ES-DOC effort more compelling to the community, if the benefits to those undertaking the extra efforts (modelling centers, MIP chairs) were more clear. As it is, ES-DOC asks those producing the data to undertake quite a lot of effort. The benefits to those wanting to analyze the data are clear; the benefits to those producing the data are less so. Here is a chance to explain why they should also embrace this effort. | We believe that the reworking of the introductory sections of this paper, setting the context of the work in terms of the workflow of running simulations in the Earth System sciences, has gone some way to make clear why the effort is useful. Our descriptions of specific instances where savings have been made to CMIP6 scientists reiterates this point. The summary makes it clear that there is a difference in the perception of the usefulness of this effort depending on how early in the workflow the engagement with ES-DOC occurs. | |

| | | | |
|---|---|---|---|
| SC2 | The authors might also take the chance to explain why undertaking the very significant effort to comply with ES-DOC requests adds value to normal scientific communications. For example by explaining how ES-DOC provides a traceable, documented answer to questions about a simulation or protocol that might otherwise need to be answered multiple times and with potentially different responses to email or other questions from analysts. | We make references to this usage in the text with phrases such as "science communication" and in the summary we talk explicitly about the provision of traceable, documented answers to questions about experiment protocols etc. | |
| SC2 | In the manuscript and as implemented, the system would be more useful if it were more flexible. One wonders, for example, what value is brought by requiring information in ES-DOC for all the solar particle forcings when the experimental descriptions do not specify them. (This seems to be the root of one of the mistakes with respect to RFMIP experiment documentation in ES-DOC.) | We believe that our system is flexible, in our paper we have been clear about the separation of our information schema from the content of that schema. The content reflects the information that the es-doc team harvested from the published CMIP6 resources. | |
| SC1 | overall the intro and sections 1 and 2 still sometimes read too much like a conversation with close colleagues in metadata management, rather than an exposition that would be widely understood by the larger community. | We have re-written the early parts of the paper, to talk more generally about the processes and challenges of running simulations in the earth system sciences. The more technical material is isolated within one section. We believe that these changes give the paper a broader appeal. | |
| RC1 | Page 1, Lines 2-3 – editorial - may communicate primarily – Change to "typically communicate". Reads better and is closer to what happens. | We have made the requested changes to the text. | typically communicate |
| RC1 | Page 1, Lines 9-10 – expected methodology – I am not sure what this means. I think it means – paths to those goals. | We have changed the text to "to aid in the inter-comparison of methodology between experiments". | to aid in the inter-comparison of methodology between experiments |
| RC1 | Page 1, Line 10 – editorial – was intended – is intended. | We have made the requested change to the text. | is intended |

| RC1 | Page 1, Line 19 – editorial clarity – Add "for MIPs" after "protocols". | This is no longer applicable as we have re-written this section of the text. | |
|---|---|---|---|
| RC1 | Page 2, Figure 1 caption – The last 2 sentences in the figure need to be moved into the text. They are too important to leave in the caption. | We think the text (in it's revised state) fits well in the caption of figure 1. Nevertheless, we have added the following to the text to the Key Concepts section: "In practice simulations can deviate in detail from the experiment protocol, that is they do not conform exactly to the requirements,..." | In practice simulations can deviate in detail from the experiment protocol, that is they do not conform exactly to the requirements |
| SC2 | Figure 1 should be changed to reflect that the responsibility and authority of defining the CMIP6 experiments is with the CMIP6-Endorsed MIPs. The experiment design and the science comes down to the MIPs as was envisioned from the beginning of the planning for CMIP6 to make it a distributed effort where the MIPs and modeling groups have ownership on the design / models, respectively. | We have reconfigured figure 1 and revised the caption. | |
| SC2 | p2, L11: "and ES-DOC use is now required for the documentation of CMIP6 simulations." please reword to "and the authors now recommended the use of ESDOC for documentation of CMIP6 simulations." | Engagement with ES-DOC is a requirement for the use of ESGF data sharing infrastructure so rather than use the text suggested by the reviewer we have changed it to "required for the documentation of CMIP6 models and ensembles". | required for the documentation of CMIP6 models and ensembles |
| SC1 | P 2 line 14: don't you mean "but also the experiments themselves" (that is, you mean ES-DOC helps not only with documenting experiments, but also with designing them)? Rephrasing the sentence would make this clearer. The logic of your "not only ...but also" here is hard to parse. | This has been addressed in our re-writing of the introductory sections of the paper. | |
| RC1 | Page 3 top – Needs a reference to figure 1a somewhere. | This has been addressed in our re-writing of the introductory sections of the paper. | |
| RC1 | Page 3, line 5 – Add reference to Eyring et al. after | We have a reference to Eyring et al 2016 in the | |

| | | | |
|---|---|---|---|
| | "DECK". | Introduction section. | |
| SC2 | p3, L10: The title of this section is misleading as the experiment are defined by the MIPs. Maybe the definition of the nomenclature? | We're interested in the structure of the definition as opposed to the scientific content. We hope that the wording now makes it clear that the subsection "Experiment Definition" is being discussed within a "The Structured Documentation" section, and that at this point it is generic and not about CMIP6 per se. | |
| RC1 | Page 3, line 14 – conform as best they can – Change to "attempt to conform". Reads better. | We have made the requested changes to the text. | attempt to conform |
| SC1 | Section 2, p. 3 - lines 15-19 - what is the Data Request coordinator, and what sort of "input" does s/he provide? By the end of the article I understand the role that "additional documentation" could play in streamlining experimental design, but it's counterintuitive at this point | We have removed the word "coordinator", since that role has not yet been defined anywhere. So this now reads: "... and the development of the data request (Juckes et al.,2019).". Later in that paragraph we say "The data request was an integral part of the process, since some MIPs were dependent on data produced in other MIPs, and in all cases the data was the key interface between the aspirations of the MIP and the community of analysts who need to deliver the science." | The data request was an integral part of the process, since some MIPs were dependent on data produced in other MIPs, and in all cases the data was the key interface between the aspirations of the MIP and the community of analysts who need to deliver the science |
| SC2 | p3, L15 "In the case of CMIP6, the iterative discussion includes input from the ES-DOC community aiming to get a formal experiment description, from the Data Request coordinator, and the CMIP6 central team at the Program for Climate Model Diagnosis and Intercomparison (PCMDI) responsible for crossexperimental common CMIP vocabularies. These extra activities result in additional documentation which can be used by those carrying out the actual experiment (figure 1b) in an attempt to minimize the burden of interpreting and carrying out many experiments." Inaccurate description of how the | We want to make clear the important distinction between the design of experiments by the CMIP6-Endorsed MIPs and the work of interpreting and clarifying them which has been done by the infrastructure teams. To that end we now say "The discussion revolved around interpreting and clarifying the MIP requirements in terms of data and experiment set up, as initially described by endorsed MIP leaders in their proposals to the CMIP panel and later in a special issue of Geophysical Model Development (GMD)." | The discussion revolved around interpreting and clarifying the MIP requirements in terms of data and experiment set up, as initially described by endorsed MIP leaders in their proposals to the CMIP panel and later in a special issue of Geophysical Model Development (GMD) |

| | | | |
|---|---|---|---|
| | experiment design in CMIP6 is established, namely by the CMIP6-Endorsed MIPs. | | |
| RC1 | Page 3, lines 30 – 31 – I note in passing that these lines make my point about the title being misleading. | Noted | |
| SC2 | P4 figure 2 needs to be changed as the workflow is not accurately representing the workflow in CMIP6. | Figures 2 is about the process by which experiments are designed in general and is not intended to be specific to CMIP6. | |
| RC1 | Page 4, line 5 – heritage – Reference needed or define. It is not clear what is meant by the word – heritage. | The text now reads: "heritage, e.g. an experiment from which initialisation fields are obtained is referred to as the parent experiment" | heritage, e.g. an experiment from which initialisation fields are obtained is referred to as the parent experiment |
| SC1 | P. 4 line 11- p 5 line 2: this paragraph seems to come out of nowhere, unattached to what precedes and follows it. Again, by the end of the article I can understand this, but not here, perhaps because it reads as if the  ˝u is about the DCPP, when it's really about something more general. | We have removed discussion of start-date ensembles, they are not referred to anywhere else in the text and we feel the section stands alone without this part. | |
| RC1 | Page 4, line 11 – editorial - shifted slightly in terms – Change to "shifted slightly from the past in terms". Clearer. | We have removed discussion of start-date ensembles, they are not referred to anywhere else in the text and we feel the section stands alone without this part. | |
| RC1 | Page 5, line 1 – Thus the DCPP . . . - It is not clear to me why this follows. Delete or make clear. | We have removed discussion of start-date ensembles, they are not referred to anywhere else in the text and we feel the section stands alone without this part. | |
| SC1 | P 5 line 8 - should be "e.g. whether the model should be ..." to avoid embedding a question in a statement. | We have made the requested change to the text. | |
| SC1 | Figure 3. Here we start to get the 0.1, 0.N, etc. notation, but it's not explained until Table 1 on the following page. | We have added the following text to the caption of figure 4: "Indices associated with the connectors indicate the numerical nature of the relationships e.g. a NumericalRequirement can have anywhere between | Indices associated with the connectors indicate the numerical nature of the relationships e.g. a |

| | | | zero to many (0.N) additional requirements whereas an EnsembleRequirement can have only one (1.1) EnsembleType." | NumericalRequirement can have anywhere between zero to many (0.N) additional requirements whereas an EnsembleRequirement can have only one (1.1) EnsembleType. |
|---|---|---|---|---|
| RC1 | Page 5, figure 3 caption – Change "intercomparison projects" to "MIPs". Why introduce new nomenclature? | | We have made the requested change to the text. | MIPs |
| SC1 | Section 2.3 - begins with a discussion of the ES-DOC controlled vocabulary, but this is a bit confusing since you have just spent 2 pages introducing your own, different vocabulary for talking about numerical experiments. Maybe simply saying "the ES-DOC controlled vocabulary" a few times early in this section would resolve the confusion. | | We have rewritten the introductory sections of the paper which now present a more general overview. Therefore we believe it makes sense to begin talking about ES-DOC concepts in what is now section 2.4. | |
| SC1 | Table 1 - probably your target audience will follow this, but I would have liked an explanation of what a "type" is. | | In the caption we describe type as "python data type". | python data type |
| RC1 | Page 7, line 2 – explicitly calling out the failure – More is needed here. Exactly what kind of information is missing? Give few examples. | | This now reads: "...explicitly calling out the failure of published papers as a medium to provide all the details of experiment requirements..." | explicitly calling out the failure of published papers as a medium to provide all the details of experiment requirements |
| RC1 | Page 7, line 2 – Add "published" before "papers". | | We have made the requested change to the text. | published papers |
| RC1 | Page 7, line 4 – 6 – This paragraph hangs. Add more or delete. If kept, explain how the present structure improves on the past in some detail and/or examples. | | We have added context to make it clear that this is in the context of work supporting scientific workflows. | |
| SC1 | P 7 lines 4-6 - this is aimed at your argument about the potential for ontologies to help with experimental design. That would work better if it flowed directly to the second half of the second sentence of the following | | We have improved the flow of this section. | |

| | | | |
|---|---|---|---|
| | paragraph, without the self-interruption about an updated ES-DOC ontology; that can go elsewhere. | | |
| RC1 | Page 7, line 9 – Add "a controlled vocabulary (CV)" before "introduced in Mattoso". | We have made the requested change to the text. | a controlled vocabulary introduced by Mattoso |
| SC1 | P 7 lines 19-21 - this is an opportunity to say more about the issue of poorly constrained experiments in the simulation sciences, which is ubiquitous and in desperate need of clear answers. I think of your work, along with the MIPs themselves, as a crucial step in the direction of that answer; this could be brought more to the fore in your article. A brief discussion of the history of MIPs, and their epistemic importance, appears at pp. C2 GMDD Interactive comment Printer-friendly version Discussion paper 349-352 of my book A Vast Machine: Computer Models, Climate Data, and the Politics of Global Warming (MIT Press, 2010). | We thank Paul for the suggestion. We feel that a complete treatment in this paper would not improve the flow of the existing material. However, we intend to take this up elsewhere. | |
| RC1 | Page 7, line 23 – I think adding "climate" before "experiments" makes things clear for the reader. | We disagree, the Soldatova and King paper is about experiment concepts in the abstract sense rather than their application in a specific field such as climate science. | |
| SC1 | P 8 line 7 - this might be better as "cannot be measured empirically," since simulations do generate (simulated) measurements. | We have made the suggested change to the text. | cannot be measured empirically |
| RC1 | Page 8, line 19 – Add "climate modeling" between "major" and "centers". | We have made the suggested change to the text. | major climate modeling centers |
| RC1 | Page 8, line 21 – driven – This is too strong. It implies the IPCC drives the process which is incorrect. The WCRP/WGCM/CMIP Panel drives the process with the IPCC timelines in view. Change "driven" to "associated | We have made the suggested change to the text. | associated with |

| | | | |
|---|---|---|---|
| | with". | | |
| RC1 | Page 8, lines 23 – 25 – Investigating differences in the models' response is missing from this list. It is the reason a 1% CO2 simulation was included and the main reason C3 GMDD Interactive comment Printer-friendly version Discussion paper for starting CMIP. The current list is very misleading. | We have made the suggested changes to the text. | to investigate differences in the models' response to increasing atmospheric CO2 |
| RC1 | Page 9, Table 3, Rationale for the DECK – Investigating the causes for differences in the models' response is missing again (see point 22 above). Again, this investigation was the main reason for starting CMIP. It continues to be important today. It is important that the rationale for these experiments be clear and accurate. | We thank Ron for this insight and have made the requested changes in the ES-DOC database from which this table is generated. | to investigate differences in the models' response to increasing atmospheric CO2 |
| RC1 | Page 9, line 6 – editorial – Change "project leaders" to "Panel". Clearer. | We have made the suggested change to the text. | Panel |
| RC1 | Page 10, lines 8 – 10 – The last sentence in this section hangs. More is needed. It needs to be clear that there are many MIPs ongoing outside of CMIP (more than 50, the last count I saw). | We have added "many" so this last sentence now begins: "There are of course many other ``non endorsed'' MIPs..." | There are of course many other ``non endorsed'' MIPs |
| SC2 | p10,l11: The title of this section is misleading. The definition of the MIPs is a scientific undertaking by communities involved in specific science questions. Perhaps a better title would be 'Documentation of the experimental design process'. | We have reframed the paper to make clear the distinction between the process of designing experiments that is captured by an ontology, the actual designing of experiments by scientists and the benefit that structured documentation can provide to those who design the experiments. It should be very clear from the additional context that we have provided that we are not claiming to have designed any of the experiments in CMIP6. | |
| RC1 | Page 10, line 12 – Add "experimental" before "design | We have made the suggested change to the text. | experiment design process |

| | | | |
|---|---|---|---|
| | process". Clearer. | | |
| SC2 | • p10,l13 "the CMIP6 team (both the CMIP panel1 and the PCMDI support group2)." This is a wrong definition of the CMIP6 team, misleading wording, please rephrase. Similarly, please also avoid the use of co-design between the CMIP Panel and PCMDI in several places of the paper. | We have rephrased this section and removed all instances of the phrase "co-design". | |
| SC2 | P11 Table 4 on the CMIP6-Endorsed MIPs should be removed. | No rationale for the removal of table 4 has been given. We included this table as an example of how the ES-DOC repository can be used to extract a complete tabular description of CMIP6 from the documentation within (and the code provided shows how this was done). It is not clear why this use-case should be excluded, and in fact we think it shows how the ES-DOC documentation could be used in future CMIP (or similar) activities to keep a dynamic and up-to-date description of agreed experiments/MIPs. Accordingly, we will leave it in unless the Editor suggests differently. | |
| SC1 | P. 12 lines 14-15 - there's something wrong with the last half of this sentence - I can't follow what's been agreed by whom, or to what. | We have made this clearer by replacing "in the GMDD paper" with "in a CMIP6-Endorsed MIP's GMD paper". | in a CMIP6-Endorsed MIP's GMD paper |
| SC1 | P 12, last 3 sentences: these could be much clearer. | We have cleaned up the description of the LUMIP "not A" etc experiments and put them in the context of forcing constraint frameworks for similar CMIP6 experiments. | |

| | | | |
|---|---|---|---|
| SC1 | Section 3.2 - p. 13 line 3: what I've really enjoyed hearing Bryan talk about is the human side of working with Metafor and similar tools (e.g. evidence of cutting and pasting in model descriptions as people become bored and inattentive during a tedious, detail-focused process). Here all of that is packed into one word - "infamous" (and a short paragraph on p 18, lines 21-25). I think it would be worth the space to discuss those experiences a bit more deeply, because they may affect modellers' ability and willingness to use the documentation for experimental design, as you're suggesting. I've attached a paper that might be of interest for your discussion of how metadata are actually used in science: Edwards, P. N., M. S. Mayernik, A. L. Batcheller, G. C. Bowker, and C. L. Borgman. 2011. "Science Friction: Data, Metadata, and Collaboration." Social Studies of Science 41 (5): 667–90. | We have added an additional sentence to cover this, but will address how we have built on lessons learned from the experiences of interacting with an excessively long and complicated CMIP5 questionnaire elsewhere. | A discussion of how the latter is done for CMIP6 (and how it builds on lessons learned from the generally poor experiences interacting with an excessively long and complicated CMIP5 questionnaire) will appear elsewhere. |
| SC1 | P 13 line 14 - G6sulphur is misspelled | We have made the suggested changes to the text. | G6Sulfur |
| SC1 | Section 4.1 is really interesting. | Thank you. | |
| RC1 | Page 14, lines 3 – 12 – forcing and temporal constraints need to be better defined. I think I understand what they are but am not sure. Some more examples would be helpful. Forcing constraints could be thought of as radiative forcing constraints, for example | We have added the following text: "An example forcing constraint might be a constraint on atmospheric composition such as a requirement for a particular concentration of atmospheric carbon dioxide." | An example forcing constraint might be a constraint on atmospheric composition such as a requirement for a particular concentration of atmospheric carbon dioxide. |
| RC1 | Page 14, line 7 – Assuming I understand things. . .add "i.e., length of simulation" after "temporal constraints". | The first phrase now reads "Temporal constraints, which specify the start date and length of simulations," | Temporal constraints, which specify the start date and length |

| | | | of simulations |
|---|---|---|---|
| RC1 | Page 14, lines 10 – 12 – I do not understand the point here. Is the point that different MIPs and simulations use differing start and end dates. Or length of simulation? If so, what is the scientific problem? Is there one? Also, it seems that these details should be documented in ES-DOCs. I assume they are and if so, this then is an issue between the authors and the MIP leaders. . ..which makes no sense to me. I am lost. | We have revised the section on temporal constraints. | |
| RC1 | Page 14 (23, 24), figures 5 and 6 – Both on my screen and in printed versions, the lines are very hard to see. The lines being hard to see means that the points make in the text are lost. | We have reproduced these figures with clearer thicker lines. | |
| RC1 | Page 15, line 27 – What is a "triples"? | We have modified the text to make it clear that ES-DOC content was used, and made the triple statement parenthetically, insofar as those who understand the Gephi tool will get benefit from knowing we use triples (and know what triples are), and those that do not, should be able to understand the meaning now. We believe a proper definition of triples would not add value to this paper. | were produced using content (in the form of triples) |
| SC1 | Section 4.3 - last sentence on p. 16 ăА˘T can you interpret this for us? What does ˇ it say about the potential for re-use of constraints, or perhaps about the particular experiments where constraints were not reusable? | There are a few forcing constraints that are used a lot, this is a good thing, we use this information to streamline the documentation burden on modelling groups. We have rewritten the section about the reuse of forcing constraints. | |

| SC1 | Section 5, Summary - last sentence on p. 17 - is this earlier involvement realistic, given the pace of change and the hectic IPCC schedule? How would it start to gain a foothold in the community? | See the section on the culture of engagement in the summary. | Early involvement of formal documentation is important for building a culture of engagement.
Our experience with the CMIP6 MIPs indicates that the process of providing detailed information about experiments was perceived in a positive way by groups when the intervention occurred early in the experiment life cycle. These groups also had a sense of ownership of their content. In contrast, groups who engaged later in the experiment life cycle were more likely to perceive the documentation effort as yet another burden. |
|---|---|---|---|
| SC2 | Section 6. We appreciate that the underlying code for ES-DOC is made publically available in a Github repository. However, when looking at the code the actual experiment description (i.e. the entries for the various experiments) seems not to be available. For traceability, it would be nice to have this all in an open Github repository as envisaged and described above. | All the documentation is and was available in an open repository. We have added a specific link to the ES-DOC documentation of the CMIP6 experiments in the "Code Availability" section. | The ES-DOC documentation of the CMIP6 experiments can be found in the ES-DOC GitHub repository at https://github.com/ES-DOC/esdoc-docs/blob/master/cmip6/experiments/spreadsheet/experiments.xlsx |
| RC1 | Page 26, rationale for the switch-on 4X simulation – The experiment does not define the equilibrium climate sensitivity (EqCS). It defines the effective climate sensitivity (EfCS). One can use the effective climate sensitivity to estimate the equilibrium climate sensitivity. See AR5 WG1 report for a discussion of this | This appeared on page 24 in the original manuscript. We have made the requested changes in the ES-DOC database from which this table is generated. | To evaluate the effective climate sensitivity of the model (EfCS) |

| | point. This is an important. Some in our community is using the two term interchangeably. This is causing problems. They are not the same thing. EfCO2 is a transient value (changes in time). EqCS is an equilibrium value, constant in time. | | |
| --- | --- | --- | --- |

[revised manuscript text omitted]

| Requirements | |
|---|---|
| **Internal Stratospheric Aerosol Precursors RCP85 to RCP45**: Injection of stratospheric sulfate aerosol precursors to reduce the radiative forcing of ScenarioMIP high forcing scenario (SSP5-85) to match that of the ScenarioMIP medium forcing scenario (SSP2-45). Modelling groups that have an internal sulfate aerosol treatment should calibrate the radiative response to sulfate aerosols individually so that the results will be internally consistent (a procedure that will be more difficult for models that have a complex microphysical treatment of aerosols). Potential methods include a double radiation call, once with and once without the stratospheric aerosols, and also the use feedback methods. Simulations to be conducted as if the aerosols or aerosol precursors are emitted in a line from 10°S to 10°N along a single longitude band (0°). The injected aerosols or aerosol precursors should be evenly spread across model layers between 18 and 20 km. Note that sedimentation processes and self-lofting due to heating are likely to result in the aerosols being distributed between 16-25 km in altitude. | **External Stratospheric Aerosol Precursors RCP85 to RCP45**: Injection of stratospheric sulfate aerosol precursors to reduce the radiative forcing of ScenarioMIP high forcing scenario (SSP5-85) to match that of the ScenarioMIP medium forcing scenario (SSP2-45). For modelling groups that have no dynamical treatment of sulfate aerosols, GeoMIP will provide a data set of aerosol optical depth, as well as ozone fields that are consistent with this aerosol distribution. Note that the amount of sulfate injection needed for a given model to achieve the goals of this experiment may vary, so modelling groups should scale the aerosol and ozone perturbation fields as necessary. Simulations to be conducted as if the aerosols or aerosol precursors are emitted in a line from 10°S to 10°N along a single longitude band (0°). The injected aerosols or aerosol precursors should be evenly spread across model layers between 18 and 20 km. Note that sedimentation processes and self-lofting due to heating are likely to result in the aerosols being distributed between 16-25 km in altitude. |
| **RCP85 Forcing**: Impose RCP8.5 forcing. *Additional Requirements:*
• Representative Concentration Pathway 8.5 Well Mixed Greenhouse Gases
• Representative Concentration Pathway 8.5 Short Lived Gas Species
• Representative Concentration Pathway 8.5 Aerosols
• Representative Concentration Pathway 8.5 Aerosol Precursors
• Representative Concentration Pathway 8.5 Land Use for Shared Socioeconomic Pathway 5 | **SSP5-85Initialisation2020**: Initialisation is from the beginning of year 2020 of the SSP5-8.5 experiment. |
| **AOGCM Configuration**: Use a coupled Atmosphere-Ocean general circulation model | **2020-2100 81yrs**: Scenario, from 2020 to the end of the 21st century |
| **SingleMember**: One ensemble member | |

**Table 8.** GeoMIP is clear about what the forcing should achieve (reduction in radiative forcing from rcp8.5 to rcp4.5) but leave it open to the modelling groups to choose a method that best suits their aerosol scheme. See https://documentation.es-doc.org/cmip6/experiments/g6sulfur for more information.

---

## Referee Report (RR1)

Reviewer Comments

Documenting numerical experiments in support of CMIP6

Pascoe et al.

Submitted to Geoscientific Model Development

Introduction:

This paper addresses an essential aspect of the use of models in scientific investigation, namely the ability to describe model workflow and experiments. This is essential for the integrity of the scientific method, the ability to design controlled experiments, and the framing of the sources of uncertainty in the experiments.

The paper addresses an effort to provide an approach across the "community," which has individuals, many levels of organization, different cultural approaches to the execution of scientific investigation, and a wide variety of disciplines and purposes.

The paper represents astoundingly difficult tasks and those tasks are likely to never be wholly complete or satisfactory. It is a task that requires sustained attention, and likely, the incorporation of concepts, if not experts, from outside of the traditional "climate modeling" community.

My recommendation is to publish the paper, essentially, as is. I am hopeful that authors will consider some of my remarks in a modified framing of the paper.

What I feel is more important than this paper or my review is a community discussion on the community approach to climate modeling and its future. It is the ability to perform this problem of documentation at scale that warrants this discussion. Is this the right way to be consolidating our resources?

Reviewer's Background

I am more of a model user than a model developer. Recently, my focus has been on the use of the CMIP models by those interested in adaptation. I have been, in the past, involved in efforts to develop improved metadata and documentation for climate models. I have worked on controlled vocabularies, and use them in my personal organization of resources. I have general experience in the focus of this paper, and a long-time participation in the climate modeling community.

Reviewer's Comments

I have read the paper, looked at some of the references, and spent some time at https://search.es-doc.org/ . Without my experience over the years, I would have difficulty entering the paper. I would appreciate more attention in the introduction about the general problems of translation and interpretation across different disciplines. There are words such as "forcing" and "scenario," which are treated as if their definitions are known and widely accepted. In the lectures I give, to scientists and model simulation users, I am always asked the question, "what do you mean by 'forcing.'" I am, presently, reviewing a very long paper defining the different uses that climate scientists attribute to scenario.

There are some very basic needs of scholarship and language that need to be recognized, for example, the need for glossaries. In every end-user design review of software systems and information services, in which I have participated, a major design gap has been glossaries. Some attention needs to be paid to these more fundamental issues of communication, rather than jumping straight into controlled vocabularies. Why the tactic of controlled vocabularies? Are there other approaches to bring order to this complex task?

Philosophically, it is perhaps worth stepping back, and looking at the nature of the problem. The community of scientists is growing, the complexity of the models is increasing, our observations of the climate are revealing many new processes that need to be considered, and the potential end user community is exploding. Hence, the idea of highly structured, stable controlled vocabulary is, perhaps, not well posed. Hence, does it make sense to consider something that is more dynamic, where there is the emergence of standards and definition?

The effort, as I read here, reminds me of a linguistic class where we discussed the efforts of the French Academy to define the French language. Language is dynamic, especially in a situation where everything is in flux, including those interested in using the language. This begs the question, do you need some expertise from other fields, such as linguists, or perhaps library science? Do you need an approach more like the Dublin Core (https://dublincore.org/ ) which looks at improving practice and, in my likely superficial interpretation, at the level of categorization that is perhaps possible and meaningful at any given time?

The paper is quite careful in defining a narrow scope of the purpose of CMIP experiments and the target audience. There is no mention of the practitioner interested in adaptation. In my opinion and experience, CMIP plays a bit of a shell game. Formally, CMIP limits its purpose to the scientific community. But it is well known that CMIP has far broader users, and there are those who proffer CMIP's broader applications. Minimally, it is worth recognizing this community and managing the expectations that will be offered by ES-DOC and its controlled vocabulary. It seems to me that this community is, ultimately, more important than the science community, and an effort is needed for translation to that community, or perhaps, we need to recognize that there is a different type of modeling approach. Followed by defining CMIP more narrowly than at present and tightly focused on its scientific mission.

If I were to approach this problem, I would seriously consider a more community based, community engaging process. I know there have been efforts in community capturing of

documentation and experience (e.g. https://journals.ametsoc.org/doi/full/10.1175/BAMS-D-14-00189.1 (CHARMe) ). I realize these approaches are problematic.  But the problems of compliance and participation that you realized in ES-DOC and CMIP5, are not likely to go away.  There is probably some "governed" space between an expert designed controlled vocabulary and reliance on the community that might offer the ability of accurate information and more completely addressing the problem of providing information for a/the "community."

Finally, I admire and congratulate the progress and successes realized in the ES-DOC effort. I, particularly, like the statements in the paper of what was learned in the CMIP5 activity and the efforts to improve it.  As I opened, this is fundamental to the scientific integrity of our field.  It needs more support, but it is not the type of activity that excites most program officers who I know. In the end, I don't think the effort can keep up with the growing complexity, and perhaps, is there a better way?